

**Climate change threatens archeologically significant ice**
**patches: insights into their age, internal structure, mass**
**balance and climate sensitivity**
**Rune Strand Ødegård[1] , Atle Nesje[2] , Ketil Isaksen[3],**
**Liss Marie Andreassen[4], Trond Eiken[5] , Margit Schwikowski[6]**
**and Chiara Uglietti[6]**
[1]{Norwegian University of Science and Technology, Gjøvik, Norway}
[2]{University of Bergen, Bergen, Norway}
[3]{Norwegian Meteorological Institute, Oslo, Norway}
[4]{Norwegian Water Resources and Energy Directorate, Oslo, Norway}
[5]{Department of Geosciences, University of Oslo, Oslo, Norway}
[6]{Paul Scherrer Institute, Villigen, Switzerland}
Correspondence to: R.S. Ødegård (rune.oedegaard@ntnu.no)
**Abstract**
Despite numerous spectacular archaeological discoveries worldwide related to melting ice
patches and the emerging field of glacial archaeology, governing processes related to ice
patch development during Holocene and their sensitivity to climate change are still largely
unexplored. Here we present new results from an extensive 6-year (2009-2015) field
experiment at Juvfonne ice patch in Jotunheimen in central southern Norway. Our results
show that the ice patch existed continuously since the late Mesolithic period. Organic-rich
layers and carbonaceous aerosols embedded in clear ice shows ages spanning from modern at
the surface to ca. 6200 BCE at the bottom. This is the oldest dating of ice in mainland
Norway. Moss mats appearing along the margin of Juvfonne in 2014 were covered by the
expanding ice patch about 2000 years ago. During the study period the mass balance record
shows a strong negative balance, and the net balance is highly asymmetric over short
distances. Snow accumulation is poorly correlated with winter precipitation and single storm





events may contribute significantly to the total winter balance. Snow accumulation is approx. 20 % higher in the frontal area compared to the upper central part of the ice patch. The thermal regime in Juvfonne is similar to what is found close to the equilibrium line of nearby glaciers. There is sufficient melt water to bring the permeable snowpack to an isothermal state within a few weeks in early summer. Below the seasonal snowpack ice temperatures are between -2 and -4°C, similar to the surrounding permafrost terrain. Juvfonne has clear ice stratification of isochronic origin. The cumulative deformation of ice over millennia explain the observed curved layering in the basal parts of the ice patch, which makes it difficult to relate the present thickness to previous thickness of the ice patch. Ice deformation and surface processes (i.e. wind and melt water) may have caused significant displacement of artefacts from their original position. Thus the dating and position of artefacts cannot be used directly to reconstruct previous ice patch extent. In the perspective of surface energy and mass balance; ice patches are in the transition zone between permafrost terrain and glaciers. Future research will need to carefully address this interaction to build reliable models.

# 1    Introduction

The emergence of glacial archaeology is described by Andrews and Mackay (2012) and Dixon et al. (2014). In archaeology, the term 'glacial archaeology' or 'snow patch archaeology' refers to several alpine contexts in different regions of the world (Callanan, 2010). The release of Ötzi's 5300 year old body from the ice marked the beginning of a number of remarkable archaeological discoveries world-wide connected to melting ice and thawing permafrost in the high mountains (Spindler, 1994). Discoveries are known from the Alps (Suter et al., 2005;Grosjean et al., 2007), mummies in Greenland (Hansen et al., 1985) and the Andes Mountains (Ceruti, 2004), and from archaeological finds at retreating ice patches in North America (Farnell et al., 2004;Dixon et al., 2005;Lee, 2012;Brunswig, 2014). When analysing the number of artefacts on a global scale during the Holocene, there is a negative correlation between periods of glacial advance and the number of artefacts. This is particularly the case in the Alps and North America (Reckin, 2013), but a similar pattern is also found in Norway (Nesje et al., 2012). The question is if this is caused by changes in climate dependent preservation conditions or decreased human use of these areas in periods of cold climate.



In Norway, there has been an increasing focus on ice patches since the extreme melting in
southern Norway in the autumn of 2006. There are about 3000 known artefact finds globally
from ice patches. Most of these have melted out during the last three decades. Approximately
2000 of these archaeological finds are in central southern Norway, making it by far the most
find-rich region in the world (Curry, 2014, pers.comm. Lars Pilø).
Among the most spectacular finds is a Bronze Age leather shoe that melted out in late autumn
2006 and a well-preserved tunic dated between 230-390 (Common Era) CE (Finstad and
Vedeler, 2008;Vedeler and Jørgensen, 2013). The shoe was dated to be around 3400 years old
(1429-1257 Before Common Era (BCE)), and is by far the oldest shoe found in Norway.
Dates are given in calibrated ages (BCE/CE) including 1 sigma errors (σ).
The geoscience of old ice patches is still in its infancy and the geoscience literature about ice
patches is sparse compared to glacial archaeology. Within the glaciological community it is
commonly differed between glaciers and snowfields and active or inactive ice (UNESCO,
1970). Snowfields may be seasonal or perennial. Seasonal snowfields melt during the
summer. Perennial snowfields exist for two years or more. Smaller ice bodies without
significant movement may be remnants of a past active glacier or a perennial snowfield and
are commonly referred to as glacierets. In this paper, we use ice patch for perennial
snowfields and glacierets. Ice patches are, in contrast to glaciers, mostly stagnant and
therefore, do not convey mass from an accumulation towards an ablation area. In fact, ice
patches often do not exhibit distinct glacier facies such as a firn area. In the wet-snow zone,
the transformation of snow to ice is fast by metamorphism and refreezing of melt water.
(Kawasaki et al., 1993). Ice patches and surrounding terrain are generally underlain by
permafrost (Haeberli et al., 2004).  There are few studies related to the thermal regime, mass
balance and dynamics (Sato et al., 1984;Fukui, 2003;Fukui and Iida, 2011;Eveland et al.,
2013). Fujita et al. (2010) concluded that they exist below the altitude of the regional
equilibrium-line altitude (ELA) of glaciers. A study by Glazirin et al. (2004) showed that they
can modify the nearby wind field. The mentioned studies have documented feedbacks
between ice patch size and both summer ablation and winter snow accumulation. The spatial
variability of the turbulent fluxes in an alpine terrain is of particular interest to ice patches. Ice
patches are influenced by advective heat transfer in summer (Essery et al., 2006;Pohl et al.,
2006;Mott et al., 2015). The sensible heat flux is reported to be to twice the net radiation input
for melting snow (Morris, 1989).



Despite some progress in these studies, the state of knowledge is not at a level to design
reliable models of how ice patches have developed during the Holocene and to evaluate their
sensitivity to future climate changes. The main objective of this study is to help fill this
knowledge gap. A multi-disciplinary approach was chosen, combining a set of new
geophysical data, radiocarbon dating, mass balance measurements and visual observations
from two 30-70 m tunnels that was excavated into the central parts of the ice patch in order to
better understand (1) the age, (2) the mass balance, (3) the thermal regime, (4) ice layering
and deformation on Holocene time scale and finally (5) the physical processes relevant to
artefact displacement and preservation.

## 11  2   Field site and physical setting

The presented research is based on a 6-year field experiment at Juvfonne ice patch, located in
Jotunheimen in central southern Norway (Fig. 1 and 2, 61.676ºN, 8.354ºE). In this area
archaeologists have so far identified more than 65 sites with finds related to ice patches, but
many sites with potential finds have not been checked in the field. The archaeological finds
are related to reindeer hunting. The snowfields are an important refuge for the reindeers
during hot summer days, giving them relief from pestering insects. Juvfonne and the
surrounding terrain is a well-preserved Iron Age hunting 'station' documented by more than
600 registered wooden artefacts and 50 hunting blinds. Radiocarbon dating of artefacts shows
ages in two separate time intervals, 246-534 CE and 804-898 CE (Nesje et al., 2012). The
geoscience studies at Juvfonne started in 2009 (Ødegård et al., 2011). Nesje et al. (2012) gave
a comprehensive presentation and discussion of archaeological finds in central southern
Norway related to Late Holocene climate history.
The width of the ice patch is approx. 500 m and upslope length 350 m. Juvfonne had an area
of 0.15 km² and ranged in altitude from 1839 to 1993 m a.s.l. in 2010 (Andreassen, 2011).
The mean surface slope is 17 degrees and the ice patch has a north-easterly aspect.
Due to snowdrift by prevailing westerly winds during the accumulation season, Juvfonne is
below the regional temperature-precipitation equilibrium-line altitude (TP-ELA). Annual
surface mass balance measurements have been conducted on three glaciers (since 1949 at
Storbreen and 1962 at Hellstugubreen and Gråsubreen) in the Jotunheimen mountain region
(Andreassen et al., 2005;Andreassen and Winswold, 2012). Except for a transient mass
surplus from 1989-1995 due to increased winter precipitation in this period, the glaciers have



lost mass. Map surveys and inventory data show a reduction in area of the glaciers in
Jotunheimen of about 10 % from the 1960s to 2003 (Andreassen et al., 2008).
Juvfonne is well within the mountain permafrost zone. Present permafrost thicknesses at
elevations where we find perennial ice patches (~> 1700 m a.s.l.) can be estimated to be more
than 100 m. Observations of ground thermal regimes (Harris et al., 2009;Farbrot et al., 2011),
bottom temperature of snow cover (BTS) (Ødegård, 1993;Isaksen et al., 2002;Farbrot et al.,
2011) and geophysical surveys to delineate the altitudinal limit of the permafrost (Hauck et
al., 2004;Isaksen et al., 2011) along with spatial numerical equilibrium and transient
permafrost models (Hipp et al., 2012;Gisnås et al., 2013;Westermann et al., 2013;Gisnås et
al., 2015)  indicate a lower limit of permafrost at 1450-1600 m a.s.l in the area.
Juvfonne is at a distance of 750 m and at the same altitude as the permafrost boreholes (the
P30 and 31 Permafrost and Climate in Europe (PACE) boreholes) and climate monitoring site
at Juvvasshøe (Sollid et al., 2000)(see Fig. 2). The site has a record of ground temperatures
and meteorological observations since September 1999. Mean annual air temperature for the
period 2000-2015 is −3.5 °C. At 15 m depth, the permafrost temperature ranges from a
minimum of -3.1 °C in 1999 to a maximum of -2.5°C recorded in 2008. The active layer
thickness has varied between 2.0 and 2.4 m and permafrost thickness is estimated to exceed
300 m (Isaksen et al., 2011). In 2008 an altitudinal transect of permafrost boreholes and
adjacent air temperature measurements were installed in the area (Farbrot et al 2013).
For the period 1961-1990 the mean annual precipitation is estimated to be between 800mm a⁻¹
and 1000mm a⁻¹ at 1900 m a.s.l. (Norwegian Meteorological Institute, unpublished data).
Results of analysis from sediment cores in the nearby Juvvatnet was used to reconstruct the
glacier activity of Kjelebrea and Vesljuvbrea (Nesje et al., 2012) following the methodology
described by Bakke et al. (2010). The results indicate that the late Holocene variations of
these glaciers are largely in agreement with size variations of other glaciers in the
Jotunheimen area (Matthews and Dresser, 2008;Nesje, 2009). Lichenometry suggests that the
margin of Juvfonne extended ~250 m from its present position during the LIA maximum
extent in the mid-18th century (Nesje et al., 2012).



## 3   Methods

### 3.1   Georadar

The ice patch was surveyed by a RAMAC georadar 23 September 2009 and 1 March 2012, using a high frequency antenna of 500 MHz. The dielectric constant of ice was set to be 3.2, giving a phase velocity of 168 m µs$^{-1}$.  Georadar data and positioning data from the Global Navigation Satellite System (GNSS) were manually digitized to obtain a point dataset of ice thickness and bed topography. The point datasets were interpolated to get an ice thickness map and a digital terrain model (DTM) of the ice patch bed. Obvious artefacts caused by the interpolation technique were manually removed. Totally 40 independent control points gave an estimated standard deviation of 1.1 m, and a maximum error of 2.6 m.

### 3.2   Laser scanning

The ice patch and surrounding terrain was scanned with an air-borne laser on 17 September 2011. The area was scanned with 5 points m$^{-2}$ with accuracy better than 0.1 m. Aerial photos were taken on the same day. These data were used to produce a high quality DTM and orthophotos of the ice patch surface and surrounding terrain. The DTM was resampled to a resolution of 1 m.

### 3.3   Mass balance and front measurements

Standard surface mass balance measurements of winter accumulation (snow depth at 20-60 sites and density at 1 site) and ablation (at 1-4 stakes) following standard methods for the melting seasons of 2010-2015 (Andreassen, 2011). Distance to the terminus has been measured from two points outside the ice patch (Fig. 3a) in August or early September using a laser distance meter.

The extent of the Juvfonne ice patch has been surveyed by foot with differential GNSS mounted on a back pack (Fig 3a, Table 1). Surveys have been done annually in August or September from 2010 to 2015, but the survey from 2012 was only done along the lower part due to snow conditions. Areal extent was also determined by digitising outlines from orthophotos from 2011 and from topographical maps from the Norwegian mapping authorities in 1981 and 2004. Furthermore, outlines from Landsat inventories from 1997 and 2003 were used (Andreassen et al., 2008;Winsvold et al., 2014). The accuracy of the





differential GNSS are within 1m, the accuracy of the N50 within 5 m and the accuracy of the
Landsat mapping within 30 m. The standard deviation in height of the GNSS measurements is
on the range 10-20 cm giving ±2 standard deviations of 0.6 m.
## 3.4 Meteorological measurements
Hourly meteorological data was obtained from the automatic weather station (AWS) at
Juvvasshøe (1894 m a.s.l.). It is the highest official meteorological station in Norway and is
freely exposed and highly representative for this study. The first station was set up in 1999
(Isaksen et al., 2003) and a new official weather station was established at the same site in
June 2009. One additional station recording hourly snow depth was set up in autumn 2011 in
front of Juvfonne (95 m from the eastern margin of the snowfield). Hourly data on snow
depth is scarce in the high mountains in Scandinavia. Observed air temperature and wind
speed on Juvvasshøe were compared against the 1971-2000 climatological normal based on
interpolated air temperature data from seNorge (Engeset et al., 2004) and daily observations
of wind speed from Fokstugu (973 m a.s.l.), 70 km NE of Juvasshøe, which was the best
nearby correlated meteorological station having long-time series.
A thermistor cable was installed in a 10 m deep borehole in 2009 to record ice temperatures.
Temperatures were recorded every 3 hours until late September 2011 with an accuracy of 0.05
℃ (1 standard deviation). The entire thermistor cable melted out in September 2014. ).
Additional thermistor measurements were made in the snow and ice at the onset of thaw in
spring 2010.
## 3.5 Radiocarbon dating
In May 2010, a 30 m long ice tunnel was excavated in the Juvfonne ice patch. During spring
2012 a new 70 metre long tunnel was excavated into the central parts of the ice patch. The
tunnels gave an excellent opportunity to verify the radar data and to collect organic material
for Accelerator Mass Spectrometry (AMS) radiocarbon dating. Dateable organic material is
available, but there are no continuous layers of organic material. Radiocarbons dating prior to
2012 are published in (Ødegård et al., 2011;Nesje et al., 2012;Zapf et al., 2013). Conventional
[14]C ages were calibrated using OxCal v4.2.4 software (Bronk Ramsey and Lee, 2013) with
the IntCal13 calibration curve (Reimer et al., 2013).



The organic debris has been collected from the walls and below the floor of the ice tunnels (5
samples from the tunnel excavated in 2010 and 5 samples from the tunnel excavated in 2012)
and organic debris melting out at the front of which two datings are reported in this paper.
Nine additional datings were published by Nesje et al. (2012).
The recently developed method for radiocarbon dating of ice utilizes the organic carbon
fraction of carbonaceous aerosols scavenged from the atmosphere during snowfall and
embedded into the ice matrix (Jenk et al., 2009;Sigl et al., 2009). This method was tested with
11 samples from Juvfonne in 2011 by comparing for the first time [14]C ages determined from
carbonaceous particles with [14]C ages conventionally obtained from organic remains found in
the ice (Zapf et al., 2013). The 2011 samples are JUV1 and JUV2 adjacent to the dated
organic-rich layers in the 2010 tunnel and a surface sample JUV3 (Table 2). In summer 2015
five samples of clear ice were collected adjacent to the plant fragment layer located just above
the bed in the tunnel excavated in 2012 (JUV0, Table 2 and 3). All blocks of ice (~20 × 15 ×
10 cm) were extracted with a pre-cleaned chainsaw and were subsequently divided into
smaller pieces. All ice blocks were transported frozen to Paul Scherrer Institute (PSI,
Switzerland), decontaminated in a cold room by removing the outer layer (0.3 mm) with a pre
cleaned stainless steel band saw and by rinsing the ice samples with ultra-pure water in a class
100 clean room (Jenk et al., 2007).
Insoluble carbonaceous particles are filtered onto preheated quartz fibre filters (Pallflex
Tissuquartz, 2500QAO-UP) and combusted with a thermo-optical organic carbon/elemental
carbon (OC/EC) analyser (Model4L, Sunset Laboratory Inc., USA), using a well-established
protocol (Swiss_4S) for OC/EC separation (Zhang et al., 2012). Analyses of [14]C were
conducted using the 200 kV compact radiocarbon system 'MICADAS' at the University of
Bern (LARA laboratory), equipped with a gas ion source coupled to the Sunset instrument,
allowing measuring [14]C directly in $CO_2$ of 3-100 µg C with an uncertainty level as low as 1%
(Ruff et al., 2010).
Dates are given in calibrated ages (BCE/CE) including 1 sigma errors (σ).





**4    Results**
**4.1    Ice thickness and ice layering**
The bed reflection was clearly seen in the radar plots (see example in Fig. 4). In addition the
ice layering was detected on most of the plots, probably due to density differences in the ice
layers (air bubbles) (Hamran et al., 2009). Georadar soundings from 2009 revealed a
maximum ice thickness of 17-19 m (Ødegård et al., 2011). The near-surface reflection
horizons are nearly parallel to the present surface. At depth, curved reflection horizons are
observed. In the ice tunnels the curved layers can be directly observed forming a distinct
angular discontinuity with the surface-parallel ice layers (Fig. 5). The surface parallel layers
have melted away since 2009 in the central and southern parts of the ice patch (Fig 6). The
DTM obtained from laser scanning combined with the bottom topography from the georadar
gave a volume of 710,000 $m^3$ in late August 2011 (mean thickness 5.6 m). The surface of
Juvfonne in September 2011 was used as the reference surface for the depth map (Fig. 3b).
The maximum depth was 16 m close to the inner part of ice tunnel excavated in 2012. In this
area the surface slope is about 18 degrees.
**4.2    Mass balance, front changes and areal extent**
Only one of the mass balance stakes (J2) existed continuously from autumn 2009 to spring
2015 (Figs. 7 and 8). Stake J2 is in the central part of the ice patch (Fig. 3a).
Snow sounding measurements (N=232) range from 0.6-4.8 m over the period 2010-2015.
Mean snow depth is 2.6 m (1.2 m w.e.). Some years show a pattern where most snow
accumulates on the leeward side of the prevailing wind the previous winter, but this is not
consistent. Inter annual variation accounts for 66%. The accumulation was further
investigated by analysing the deviation from mean each year. This dataset contains a
significant trend with increased accumulation towards the front (Fig. 3c and Table 4). The
difference between the upper central area and the front is 0.2 m w.e (Fig 3c), which
corresponds to approx. 20% increase in accumulation.
The total mass loss is measured to 10 m of ice at the site of the thermistor measurements (Fig.
3a). The 10-metre thermistor cable installed on the 29th of October 2009 melted out in mid-
September 2014. The total mass loss at stake J2 was 10.5 m w.e. during the same period.



Elevation changes from September 2011 to September 2014 are shown in Fig. 3d. These
results are based on the laser scanning in 2011 and differential GNSS-tracking in 2014. The
measurements show a highly significant asymmetric pattern with close to zero surface
elevation changes in the western part and surface lowering of 3-5 m in the eastern and central
part of the ice patch. This strong gradient is measured over distance of just 200 m at
approximately the same altitude. The part with most negative change has more than average
accumulation.
Front change measurements were initiated in 2009 at JF1 and in 2010 at JF2 (Fig. 9). The
measurements revealed that Juvfonne retreated in all years except in 2012 and 2015 where the
ice patch increased its size due to excessive snow that formed a thin ice and snow layer
around the margin. The total retreat 2009-2014 is -52 m measured from JF1 and over 2010-
2014 the mean change is 44 m (-51m from JF1 and -38 m from JF2).
The annual extent measurements (2010-2015) show area fluctuations of the margin, varying
from 0.101 km$^2$ (9 September 2014) to a maximum of 0.186 km$^2$ on 11 September 2015
(Table 1). The extent measurements show that the ice patch shrinks and grows along the
whole margin.
**4.3   Climate parameters**
Air temperature and wind speed at Juvvasshøe for the period 2000-2015 are outlined in Fig.
10 a-b over the ablation season (June-September). The mean June-September air temperature
in this period is 3.2 °C (1.0 °C above the 1971-2000 mean). Air temperatures, near-ground
surface temperatures and frequency of days with daily mean air temperature above 0 °C (the
two latter are not shown in Fig. 10) are high in summers 2002, 2003, 2006, 2011 and 2014,
and especially 2006. Observations from nearby weather stations with long climate series
reported record-breaking temperatures in late summer and autumn 2006. In the investigation
period 2009-2015 the coldest summer was 2012, which was the only summer below the 1971-
2000 mean (Fig. 10).
In general there is a high frequency (35-58 days per season) of strong breeze during the period
2009-2015 (Fig. 10b). Comparing wind data from the AWS at Fokstugu indicates two to three
times more frequent strong wind than 1971-2000 mean during the investigation period.
Observed incoming short- and longwave radiation from Juvvasshøe (not shown) show no





clear patterns related to single summers, but 2011 peaks out as the summer with greatest
incoming longwave radiation.
For snow accumulation or abrasion on ice patches wind speed and wind direction is crucial.
Results reveal (not shown here) that strong wind is frequent during winter. There are great
variations from year to year and between early and late winter in respect to frequency of
strong gale and wind direction.

## 8    4.4   Snow measurements and modelling

The automatic snow depth observations in front of Juvfonne show great hourly to daily
variability and there is distinct different pattern of snow accumulation between the four winter
seasons (Fig. 11). The greatest increase in snow depth during early and mid-winter in all years
is related to storm events. This is also the case for strong snow depth decrease events (mainly
due to abrasion). Comparing the observed and modeled snow depths (which not take into
account redistribution of snow by wind), it is clear that much of the accumulation is not
correlated with precipitation (Fig. 11). The modelled snow depth for Juvfonne was obtained
from a precipitation/degree-day model operating on $1 \times 1$ km$^2$ developed for a web-based
system (http://senorge.no/) for producing daily snow maps for Norway (Engeset et al.,
2004;Saloranto, 2012).  A similar poor correlation ($r^2$=0.24) is also found for very small
glaciers in the Alps (Huss and Fisher, 2016)
The observed melt in central parts (J2) was compared with a degree-day model using typical
values calculated from nearby glaciers (Fig. 7) (Laumann and Reeh, 1993). This modelling
shows a quite good fit except the 2010 season. In this season the summer balance was about
twice the outcome of the degree-day model.

## 25   4.5   Temperature of ice and permafrost

Temperature measurements in Juvfonne reveal 10-m depth ice temperature in the range of -2
to -4 ºC (Fig. 12). The ice and snow temperature results show that the Juvfonne ice patch is
cold-based and underlain by permafrost (Fig. 13). The measurements at 5-10 m depth in the
ice are similar to the measurements in the nearby permafrost borehole at Juvvasshøe (Fig. 12).



In spring, the melt water percolates and refreezes in the snowpack until the snow is isothermal
at a temperature close to 0ºC. The surface melt water does not percolate through the level of
the winter cold wave. The heat flow into the ice is gradually decreasing during the melt
season. Superimposed ice forms at the level of impermeable ice.

## 6   4.6   Radiocarbon dating

The AMS radiocarbon dating obtained from organic-rich layers and from carbonaceous
aerosols embedded in clear ice in the Juvfonne ice patch shows ages spanning from modern at
the surface to ca. 6200 BCE at the bottom (clear ice below the basal organic-rich layer), thus
showing that Juvfonne has existed continuously during the last ~7500 yrs. So far, the basal ice
in Juvfonne is the oldest dated ice in mainland Norway (Table 2).
In the tunnel opened in 2010 the AMS radiocarbon dating of organic matter embedded in the
ice shows modern age in the top layer at the entrance, and ages ranging from 1218-1125 BCE
to 945-987 CE inside the tunnel. These results were previously published in Nesje et al.
(2012) and recalibrated for this study (Fig. 14).
In the tunnel opened in 2012 the AMS radiocarbon dating of five organic layers embedded in
the ice about 70 m from the margin of the ice patch, yielded dates in chronological order from
the base upwards, ranging from 4711-4606 BCE at the base to 53 BCE – 21 CE in the ceiling
of the ice tunnel, approximately 2.5 m above the tunnel floor. The organic debris that yielded
the oldest age was collected from the innermost part of the ice tunnel, about 0.3 m above the
bed. The layer where the sample was retrieved could be followed close to the bed in the inner
parts of the tunnel. The carbon dates on carbonaceous aerosols were sampled at the same
location to the side and below the plant fragment layer (Table 3). The oldest dating is 6418-
5988 BCE. The position of the sample site is marked on Fig. 3a.
In the autumn 2014, two *in-situ Polytrichum* moss mats melted out along the margin of
Juvfonne south of the ice tunnel excavated in 2010. AMS radiocarbon dates of the two moss
mats indicate that the moss was killed by the expanding margin of the ice patch about 2000
years ago (1 BCE -54 CE – Poz-66166 and 5-68 CE – Poz-66167). Thus the minimum extent
of the south-eastern part of the ice patch observed in September 2014 is most likely the
smallest in 2000 years.





With the exception of one identified outlier, the obtained results from dating of carbonaceous
aerosol particles in the ice could reproduce the expected ages very well (Zapf et al., 2013).
This gives confidence that the age of organic debris in the ice is similar to the surrounding ice.
**5   Discussion**
The discussion focuses on the value of this research in the context of the long-term objective
to develop models of mass balance and thermal regime on Holocene time scale at ice patches
and surrounding terrain.
The discussion is organised in four sections: (1) the mass balance, (2) thermal regime, (3) ice
layering and deformation on Holocene time scale and (4) the environmental processes
relevant to artefact displacement and preservation.
**5.1   The mass balance**
Perennial ice patches are, due to their existence, located at sites with close to long-term zero
mass balance. The inter-annual variability in summer and winter balance could be
considerable, but the long-term changes in mass must be close to zero as long as they do not
disappear or develop into a glacier. The 6-year record of mass balance gives some insight into
the spatial and temporal variability of the mass balance.
The snow accumulation during the 6-year period (2010-2015) shows increased accumulation
towards the front of the ice patch. This is probably a response to increased melt. Along the
outer rim of Juvfonne the surface altitude changes (negative net balance) vary between less
than 1m to nearly 5m within a 200 m distance at same altitude over a period of 3 years (Fig.
3d). Field data is consistent with the interpretation of increased melting due to sensible and
latent heat fluxes. Micro-meteorological investigations by Mott et al. (2011) of processes
driving snow ablation in an alpine catchment show that advection of sensible heat cause
locally increased ablation rates at the upwind edges of the snow patches.
The 2010 anomaly in the summer balance is most likely related to increased melt during
periods with strong south and south-easterly winds (unsheltered direction for Juvfonne)
combined with relatively high air temperatures and high relative humidity causing enhanced
turbulent fluxes. This 2010 anomaly is probably the reason for the asymmetric net balance of
Juvfonne (Fig. 6). Exceptionally large melt episodes have recently been reported from the



southern and western part of Greenland ice sheet in July 2012, where nonradiative energy
fluxes (sensible, latent, rain, and subsurface collectively) dominated the ablation area surface
energy budget during multiday episodes (Fausto et al., 2016).
The snow recording from the station in front of Juvfonne (95 m from the front) clearly
illustrates the complexity of snow accumulation in this environment In front of Juvfonne
abrupt changes in snow depth within hours dominate the series, causing great day-to-day
variability. These changes seem to be mainly driven by the rate of wind speed and wind
direction. One single storm events with westerly winds could account for almost 50% of the
winter accumulation in less than 24 hours (Figure 11, 2014-15). Spring snow accumulation
with insignificant wind drift could also influence mass balance, like the 2012 season.

## 11   5.2   Ground and ice thermal regime

Juvfonne consists of cold ice surrounded by permafrost terrain (Fig. 12). Perennial ice patches
can be used as indicators of local (mountain) permafrost conditions. The physical background
is that their ice cannot warm above 0°C in summer, but cool down far below 0°C during the
cold season. Holocene permafrost modelling (Lilleøren et al., 2012) suggest that permafrost
survived the highest areas of the Scandinavian mountains during the Holocene thermal
maximum (HTM), and thus permafrost ice could be of Pleistocene age. Radiocarbon dates
from Juvfonne show that the deepest central part of the ice patch contains carbonaceous
particles embedded in the ice 6418-5988 BCE (JUV0_5 - Table 2). This is a strong indication
that Juvfonne has existed continuously since mid-Holocene, and the dating of the ice could
offer strongly needed validation of Holocene permafrost models. Juvfonne could contain
older ice, and it is most likely that ice patches at higher altitude contains older ice.
The thermal regime of the ice in Juvfonne is similar to what is found close to the equilibrium
line of nearby glaciers (Sørdal, 2013;Tachon, 2015). The temperature measurements show
that there is sufficient melt water to bring the permeable snowpack to an isothermal condition
within a few weeks in early summer (Fig. 13). Below the seasonal snowpack, the ice remains
cold during the summer with temperatures on the range -2 - -4°C at 5-10 m depth (Fig. 12).

## 28   5.3   Ice layering and deformation on Holocene time scale

In the central parts of the ice patch, a first order estimate of maximum basal shear stress is on
the range of 30-45 kPa (no averaging of surface slope). Adding 5 m to the depth will increase





the shear stress to 40-60 kPa for the central part. The latter is probably close to the range for the last decades. The cumulative deformation of ice (maximum ~30-60 kPa basal shear stresses) over millennia explains the observed curved layering in the basal layer of the ice patch (Fig. 4). Cumulative ice deformation on a time scale of several millennia makes it difficult to relate the present thickness and slope of theses layer to previous thickness of the ice patch.

The observed ice layers almost certainly represent surface of isochronic deposition. Within both ice tunnels in Juvfonne there are several organic/debris layers of uncertain origin. From the appearance of these layers, it is probably wind or water transported material or reindeer droppings. In the case of Juvfonne, there is a reasonable correlation between the age of the ice and the age of the organic layers (Zapf et al., 2013). This is necessarily not the case at other ice patches.

The empirical relation between basal shear stress and altitude range of glaciers was investigated by Haeberli and Hoelzle (1995) based on data from the European Alps. A basal shear stress of 15-20 kPa is in good agreement with the values for ice bodies with elevation ranges of 150m as at Juvfonne.

## 5.4 Artefact displacement and preservation

From a cultural management perspective, there is particular interest in developing methods to identify sites of interest (Rogers et al., 2014) and a better understanding of the environmental treats (Callanan, 2015). The environmental treats are mainly related to subaerial exposure of artefacts. Especially leather artefacts, textiles and steering feathers of arrows are exposed to movement and decomposition short time after melt out. Wooden objects are more resistant.

The artefacts at Juvfonne have been found in permafrost terrain surrounding the ice patch, most of them are found in the front of the ice patch within a few tens of meters of the ice patch. The wooden artefacts range from 250-900 CE. Even during the extreme minimum in September 2014 (Fig. 6) there are no observations of artefacts melting out within the ice.

The exposure time to physical processes and microbial activity is critical to artefact decomposition. At Juvfonne there is a gradual increase in the ground exposure time depending on snow accumulation and melt over millennia. The oldest ice found so far is





6418-5988 BCE (JUV0_5 - Table 2). At the eastern edge AMS radiocarbon dates show that
the moss mats were covered (killed) by the expanding snowfield about 2000 years ago (Table
2, Poz-56952). Lichenometry indicates that the front of Juvfonne extended ~250 m from its
present position during the LIA maximum in the mid-18th century (Nesje et al., 2012). A
photo of Juvfonne from around 1900 shows the front close to the expected LIA extent. These
results constrain the extent of the ice patch since the mid-Holocene, but temporal and spatial
variability need to be considered to assess the actual exposure time of artefacts.
Several radiocarbon dates of the top layer in 2010 (Fig. 4) show modern age. This means that
artefacts found at Juvfonne have been sub-aerially exposed after the LIA but prior to 2009.
Thus the dating and position of artefacts cannot be used directly to reconstruct previous ice
patch extent.
Juvfonne and surrounding terrain is an active environment in terms of geomorphological
processes. In particular, during the extreme melting in autumn 2014 several small
accumulations of organic material/debris occurred at the upper margin of the ice patch.
Within a few days, melt water moved this material to the front of the ice patch. Downslope
movement of artefacts by melt water is certainly possible at Juvfonne. Finds at other ice
patches in Jotunheimen supports this interpretation, where different pieces of the same
artefact have been found along the direction of steepest slope. Textiles and leather objects are
more likely transported by wind, and preservation at its original position is less likely. There
are no finds of textiles or leather objects at Juvfonne.
**6   Conclusion**
Based on a 6-year field experiment on Juvfonne ice patch in central southern Norway, the
following main conclusion could be drawn:
• Ice stratigraphic characteristics and radiocarbon dating strongly suggest that the
Juvfonne ice patch was small or absent during Holocene thermal maximum, but
existed continuously since ca. 6200 BCE (the late Mesolithic period) without
disappearing or developing into a glacier with basal sliding. The oldest radiocarbon
dates show that the deepest central part of the ice patch contains carbonaceous




particles embedded in the ice 6418-5988 BCE, which is the oldest dating of ice in
mainland Norway.
• Radiocarbon dates show that the moss mats appearing in 2014 were covered (killed)
by the expanding snowfield about 2000 years ago. The minimum extent observed in
September 2014 at the south-eastern part is most likely the smallest ice patch in ~2000
years.
• A 6-year record of mass balance measurements shows a strong negative balance. The
total mass loss at one site was 10.5 m w.e. Elevation changes are highly asymmetric
over short distances, from close to zero to surface lowering of several meters. There is
a significant increase in snow accumulation towards the front of approx. 20%
compared to the upper central area. Assuming that this is a close to equilibrium
situation, increased accumulation reflects increased melt. Locally increased ablation
rates are probably caused by significant spatial variability of the sensible and latent
heat fluxes. The melt anomaly in 2010 is most likely related to periods of strong
south-easterly winds and high relative moisture boosting the turbulent fluxes.
• The winter balance is poorly correlated with winter precipitation. One single storm
events may contribute significantly to the winter balance.
• The thermal regime of the ice in Juvfonne is similar to what is found close to the
equilibrium line of nearby glaciers. Temperature measurements show that there is
sufficient melt water to bring the permeable snowpack to an isothermal state within a
few weeks in early summer. Below the seasonal snowpack, at 5-10 m depth, the ice
remains cold with temperatures between -2 and -4°C. The cold ice is surrounded by
permafrost terrain having similar ground temperatures.
• Geophysical investigations show a clear stratification. The observed ice layers almost
certainly represent surface of isochronic deposition. At depth, curved reflection
horizons are observed consistent with cumulative ice deformation over millennia.
Even a thin ice patch like Juvfonne (<20 m thick) ice deformation is a critical factor in
the interpretation of the ice layering and makes it difficult to relate the present
thickness and slope of theses layer to previous thickness of the ice patch.
• Ice deformation and surface processes (i.e. wind and melt water) may have caused
significant displacement of artefacts from their original position.





•   Artefacts melted out in front of Juvfonne since 2009 have been sub-aerially exposed
2       after the LIA but prior to 2009. Thus the dating and position of artefacts cannot be
3       used directly to reconstruct previous ice patch extent.

The exploratory analyses of field data from Juvfonne show for the first time the geoscience
research potential of ice patches in Scandinavia. The results give new insights into their age,
internal structure, mass balance and climate sensitivity, and have taken the state of knowledge
to level where models can be designed. The feedback mechanisms observed on Juvfonne
suggest that ice patches are robust to climate change, at least on the time scale of decades.
Perennial ice patches are, due to their existence, areas with close to long-term zero mass
balance. However, they are probably more sensitive than glaciers to changes in the wind
pattern. In the perspective of surface energy and mass balance; ice patches are in the transition
zone between permafrost terrain and glaciers. Future research will need to carefully address
this interaction to build reliable models of how ice patches have developed during the
Holocene and their response to future climate change.
**Acknowledgements**
We thank the archaeologists Lars Pilø and Espen Finstad for valuable comments and
discussions related to artefact displacements and Dag Inge Bakke at Mimisbrunnr Klimapark
2469 for support in the field. Professor Emeritus Wilfried Haeberli and Professor Bernd
Etzelmüller gave useful comments to an earlier version of the manuscript and are gratefully
acknowledged.



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

results from a case study in Jotunheimen, southern Norway, Geophysical Research Abstracts
Vol. 13, EGU2011-12027, EGU General Assembly 2011, Vienna, 2011, 1,





Table 1.
Areal extents of Juvfonne derived from topographic maps, Landsat imagery, GNSS
measurements by foot and digitising from orthohotos. *Seasonal snow remaining along the
extent.

| Year | Date | Source | Area (km$^2$) |
|---|---|---|---|
| 1981 | | map | 0.171 |
| 1984 | 10.08.1984 | Orthophoto | 0.208 |
| 1997 | 15.08.1997 | Landsat | 0.208 |
| 2003 | 09.08.2013 | Landsat | 0.150 |
| 2004 | 12.08.2004 | map | 0.187 |
| 2010 | 25.08.2010 | GNSS | 0.149 |
| 2011 | 02.08.2011 | GNSS | 0.150 |
| 2011 | 17.09.2011 | Orthophoto | 0.127 |
| 2012 | 12.09.2012 | GNSS | 0.160 |
| 2013 | 12.08.2013 | GNSS | 0.151 |
| 2014 | 09.09.2014 | GNSS | 0.101 |
| 2015 | 11.09.2015 | GNSS | 0.186* |



Table 2. AMS radiocarbon dates from the ice tunnels and ice samples from ice patch surface.

**Ice tunnel 1 (opened 2010)**

|  |  |  |  | Calibrated ages | |
|---|---|---|---|---|---|
| Lab. no. | Dated material | Radiocarbon age BP | Median probability | 1 sigma (68.3%) | 2 sigma (95.4%) |
| Poz-37877 | Organic remains | 1095 ± 30 | CE 949 | CE 945-987 | CE 890-1012 |
| Poz-37879 | Organic remains | 1420 ± 30 | CE 627 | CE 612-651 | CE 582-660 |
| Poz-39788 | Reindeer dung | 1480 ± 30 | CE 586 | CE 557-614 | CE 539-644 |
| Poz-37878 | Organic remains | 1535 ± 30 | CE 511 | CE 532-569 | CE 428-592 |
| Poz-36460 | Organic remains | 2960 ± 30 | BCE 1172 | BCE 1218-1125 | BCE 1262-1072 |

.................................................................................................................................

Radiocarbon dates on carbonaceous aerosols trapped in the 'clean' ice matrix (Paul Scherrer Institute, Villigen, Switzerland; Zapf et al., 2013: Radiocarbon 55 (2-3), 571-578)

|  |  | Calibrated ages BP (=AD 1950) | | |
|---|---|---|---|---|
| Lab. no. | Radiocarbon age | Median probability | 1 sigma (68.3%) | Comments |
| JUV3_1 42845.1 | -940 ± 95 BP | -42 BP (1950) | -42 - -47 BP | Sample from ice patch surface |
| JUV3_2 42845.2 | -723 ± 113 BP | -48 BP | -46 - -53 BP | Sample from ice patch surface |
| JUV3_3 42845.3 | -1157 ± 102 BP | -36 BP | -8 - -42 BP | Sample from ice patch surface |
| JUV3_4 42845.4 | -1221 ± 116 BP | -34 BP | -8 - -41 BP | Sample from ice patch surface |
| JUV-3 2010 mean | -1010 ± 107 BP | -41 BP | -40 - -45 BP (Modern) | Samples from ice patch surface |

|  |  |  | Calibrated ages | |
|---|---|---|---|---|
| Lab. no. | Radiocarbon age BP | Median probability | 1 sigma (68.3%) | 2 sigma (95.4%) |
| JUV2_1 43443.1 | 1021 ± 205 | CE 995 | CE 861-1211 | CE 640-1312 |
| JUV2_2 43443.2 | 1874 ± 665 | CE 45 | BCE 592-CE 724 | BCE 1433-CE 1329 |
| JUV2_3 43443.3 | 1121 ± 321 | CE 891 | CE 640-1221 | CE 251-1432 |
| JUV2_4 43443.4 | 1126 ± 284 | CE 892 | CE 652-1169 | CE 381-1405 |
| JUV2 2010 Mean | 1286 ± 409 | CE 720 | CE 378-1165 | BCE 169-1442 |
|  |  |  |  |  |
| JUV1_1/2 43442.1 | 3875 ± 342 | BCE 2353 | BCE 2776-1924 | BCE 3138-1501 |
| JUV1_3 43442.2 | 2144 ± 303 | BCE 200 | BCE 541-CE 172 | BCE 846-CE 475 |
| JUV1_4 43442.3 | 2647 ± 711 | BCE 834 | BCE 1641-CE 69 | BCE 2588-CE 775 |
| JUV1 2010 Mean | 2889 ± 488 | BCE 1105 | BCE 1643-471 | BCE 2346-CE 85 |

**Ice tunnel 2 (opened 2012)**

|  |  |  |  | Calibrated ages | |
|---|---|---|---|---|---|
| Lab. no. | Dated material | Radiocarbon age BP | Median probability | 1 sigma (68.3%) | 2 sigma (95.4%) |
| Poz-56952 | Organic remains | 2025 ± 30 | BCE 25 | BCE 53-BCE 21 | BCE 111-CE 55 |
| Poz-56953 | Organic remains | 3490 ± 35 | BCE 1816 | BCE 1831-1767 | BCE 1904-1737 |
| Poz-56954 | Organic remains | 4595 ± 35 | BCE 3367 | BCE 3376-3340 | BCE 3382-3326 |
| Tra-4427 | Organic remains | 5044 ± 100 | BCE 3841 | BCE 3954-3761 | BCE 4001-3645 |
| Poz-56955 | Organic remains | 5800 ± 40 | BCE 4651 | BCE 4711-4606 | BCE 4729-4544 |

.................................................................................................................................

Radiocarbon dates on carbonaceous aerosols trapped in the 'clean' ice matrix sampled and dated in 2015 (Paul Scherrer Institute, Villigen, Switzerland)

|  |  |  | Calibrated ages | |
|---|---|---|---|---|
| Lab. no. | Radiocarbon age BP | Median probability | 1 sigma (68.3%) | 2 sigma (95.4%) |
| juv1-1 4184.1.1 | 5909 ± 248 | BCE 4807 | BCE 5061-4495 | BCE 5373-4319 |
| juv1-2 4380.1.1 | 6300 ± 138 | BCE 5255 | BCE 5384-5203 | BCE 5525-4932 |
| juv2-1 4185.1.1 | 6521 ± 217 | BCE 5455 | BCE 5664-5292 | BCE 5877-4985 |
| juv2-2 4381.1.1 | 6565 ± 135 | BCE 5514 | BCE 5628-5463 | BCE 5730-5293 |
| juv3-1 4186.1.1 | 7306 ± 232 | BCE 6178 | BCE 6418-5988 | BCE 6602-5725 |
| juv3-2 4382.1.1 | 6682 ± 227 | BCE 5609 | BCE 5812-5463 | BCE 6049-5207 |
| juv5-1 4187.1.1 | 7293 ± 219 | BCE 6166 | BCE 6397-5987 | BCE 6532-5734 |
| juv5-2 4383.1.1 | 6405 ± 230 | BCE 5336 | BCE 5564-5204 | BCE 5735-4800 |
| juv 0 (2015) Mean | 6623 ± 210 | BCE 5555 | BCE 5733-5359 | BCE 5983-5206 |





1    Table 3.

| Sample blocks | Sample description |
|---|---|
| 1 | JUV 0_1 and JUV 0_2:  the side of the ice step with plant fragment layer. Clear ice divided into two sub samples. |
|  | Since there was no place to cut off further ice, the other samples were taken from the wall on the left side of the corner where the ice step is located. |
| 2 | JUV 0_3 and JUV 0_4: divided into two subsamples. This sample broke into pieces during cutting, but it is clear ice. |
| 3 | JUV 0_5 and JUV 0_6: nice and clear ice block cut at the right of sample 4. It was divided into two subsamples. |
| 4 | This ice block contains a lot of dark organic material. For the moment it is stored in the cold room and has not been processed. It could be measured with the conventional radiocarbon procedure and it is possible to separate some clear ice for the carbonaceous dating approach. |
| 5 | JUV 0_7 and JUV 0_8: clear ice cut inside the hole left after cutting sample 3. It was divided intp two subsamples. |



1    Table 4

2    Key statistics for the first order polynomial fit of snow accumulation (deviation from mean

3    each year) in the period 2010-2015.

| Sources of variation | Sums of squares | Degrees of freedom | Mean Square | F-test |
|---|---|---|---|---|
| First order polynomial regression | 0.656 | 2 | 0.328 | 6.339 |
| Deviation | 11.847 | 232-2-1 (229) | 0.052 | |
| Total variation | 12.503 | 232-1 (231) | 0.054 | |



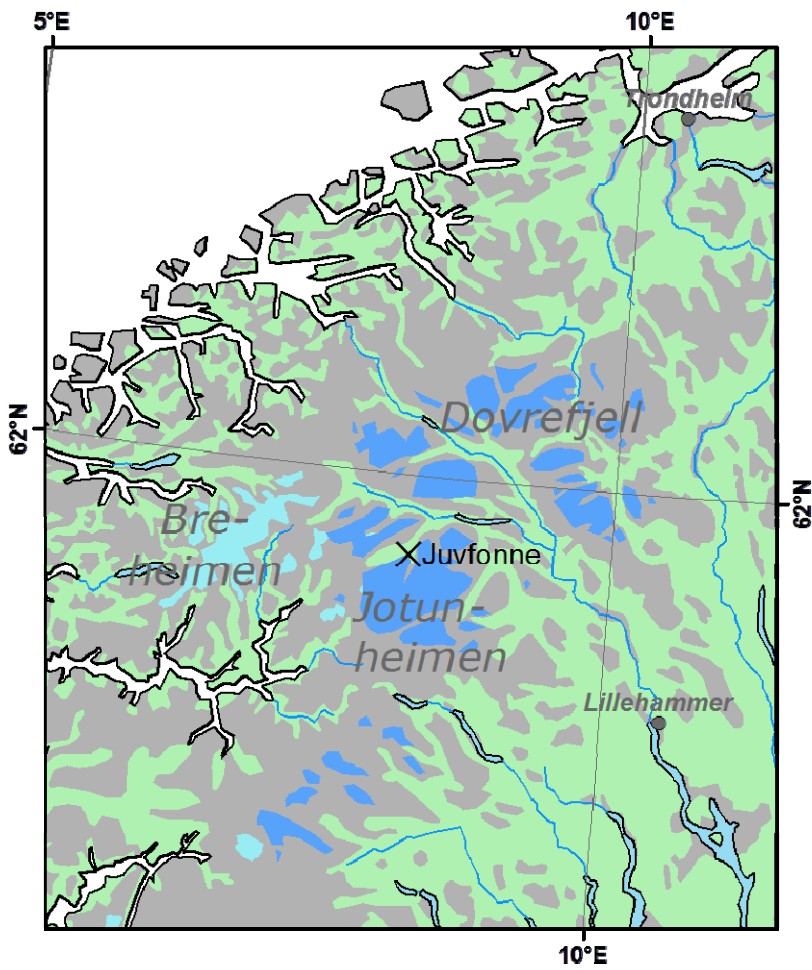

3    Figure 1. The field site Juvfonne (marked with X) in central southern Norway. Dark blue is

4    permafrost areas, light blue is glaciers. Permafrost extent generalized from Lilleøren et al.

5    (2012).





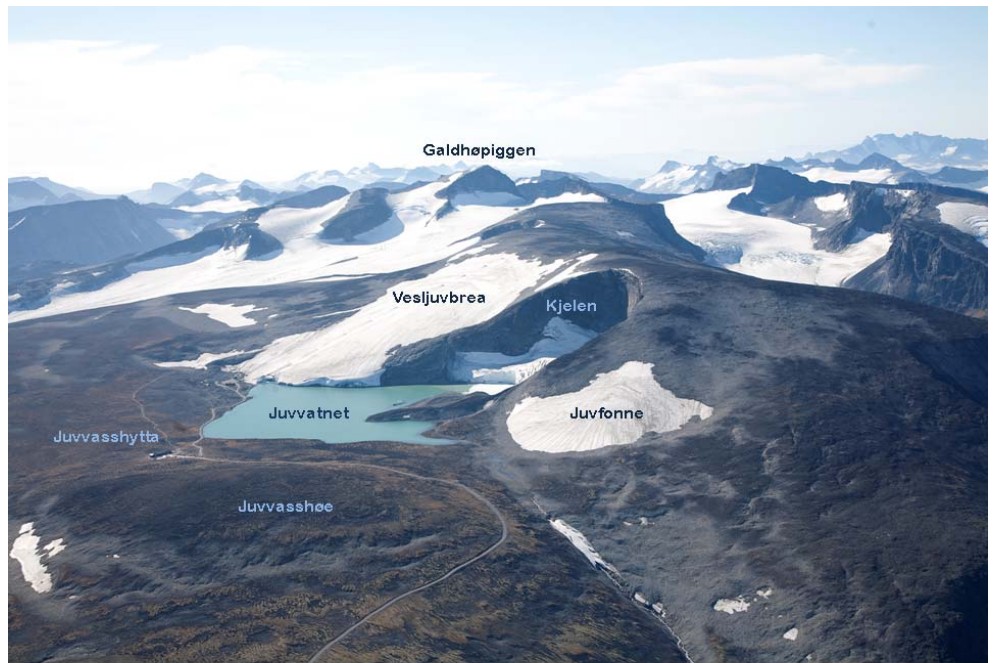

Figure 2. Overview picture of Juvfonne and the Juvflye area including Kjelen. Juvvatnet,
Juvvasshytta, Vesljuvbreen and the P30 and 31 Permafrost and Climate in Europe (PACE)
boreholes at Juvvasshøe. Also visible is the highest mountain of Norway, Galdhøpiggen
(2469 m a.s.l.). Photo: Helge J. Standal.



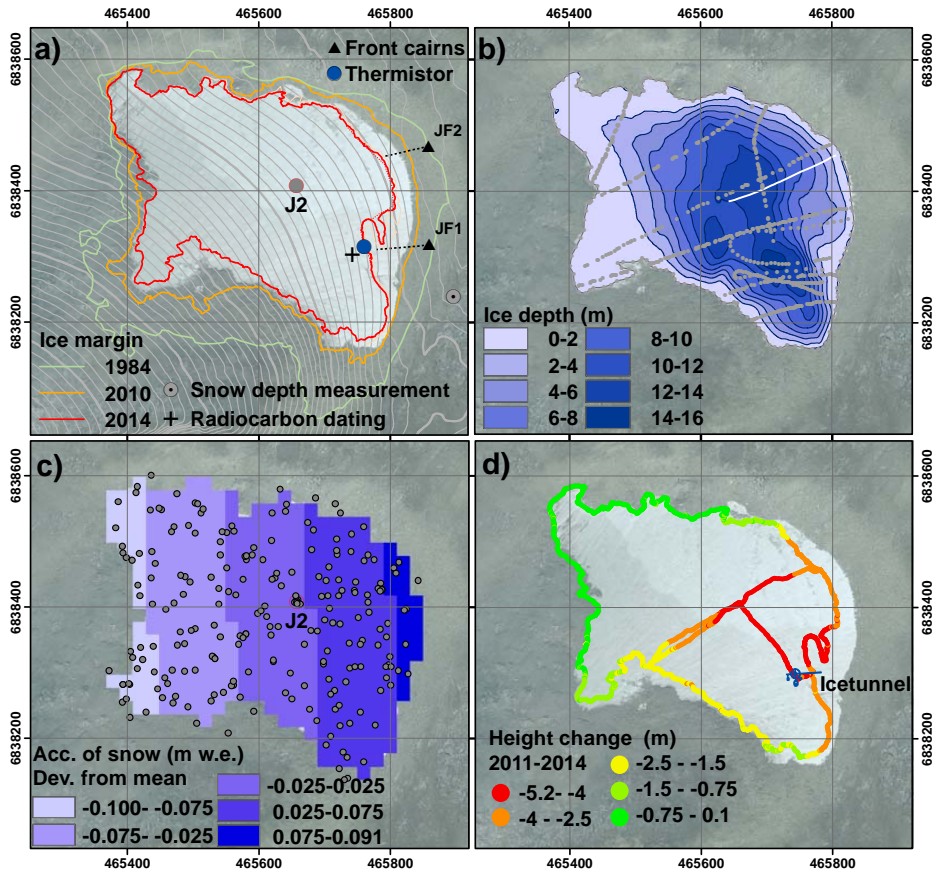

Figure 3. Maps of Juvfonne with ortofoto from September 2011 as background, a) ice
margins, position of front measurements (JF1 and JF2- see figure 14), position of mass
balance stake J2, position of thermistor for ice temperature measurements (Fig. 12) and
position of the oldest radiocarbon dating and position of snow depth measurement station, b)
interpolated contours of bed topography relative to ice thickness in September 2011 (grey
markers are radarpoints used in the interpolation) and position of the georadar track in Fig. 4 -
white line, c) grey markers are snow depth measurements (2010-2015), the raster map shows
a first order polynomial fit to the deviation from mean accumulation each year (see table 3 for
details) d) height differences along GNSS tracks in 2014 relative to ice surface from laserdata
in 2011 and positions of ice tunnel excavated in 2012.





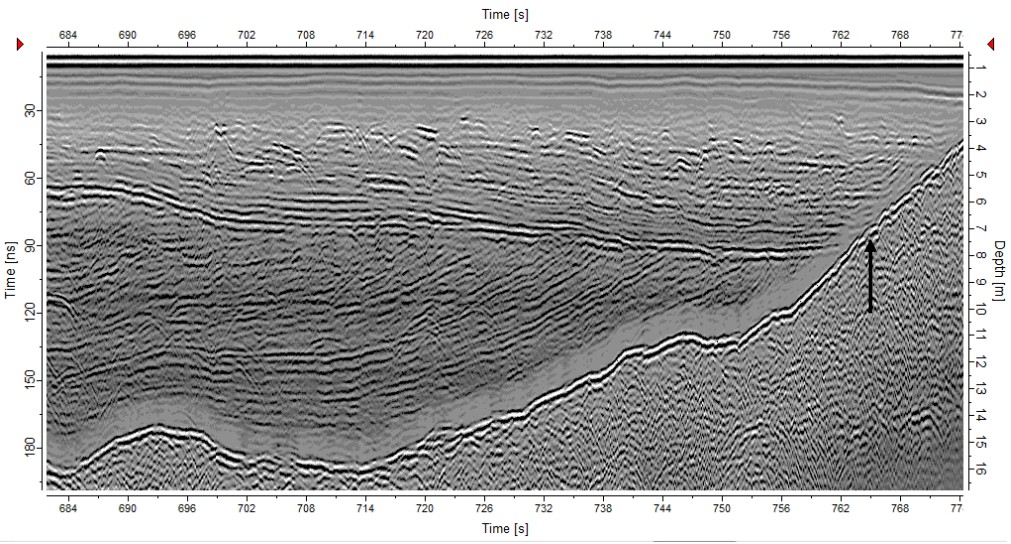

3    Figure 4. Example of 500 MHz Georadar profile. The position of the track shown in figure

4    3b. The arrow shows minimum front position in September 2014 (Ice velocity: 168 m µs-1,

5    adjustment velocity: 300 m µs-1, automatic gain control, scale factor 5000).





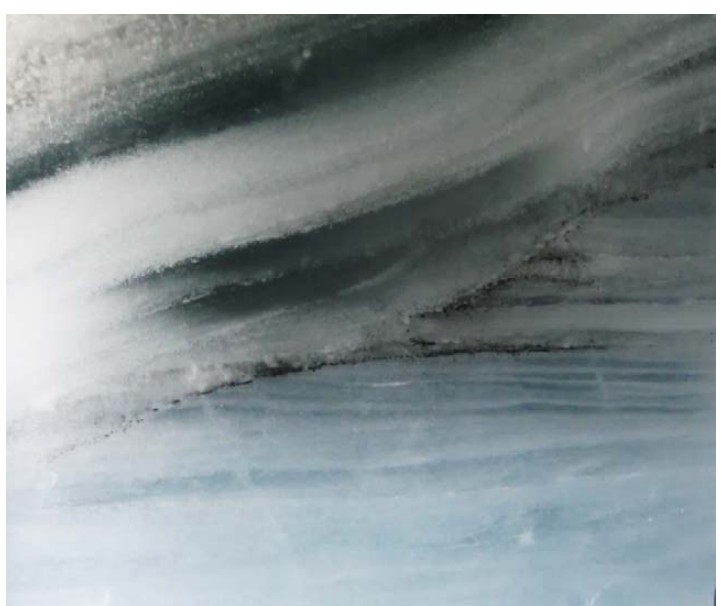

Figure 5. Photo of angular discontinuity at the wall of the 2010 ice tunnel, as also observed on
the georadar data (Fig. 4). The upper layering is parallel to the surface of Juvfonne.
Radiocarbon dating of the upper part showed modern age. Width of picture is approximately
0.4 m.



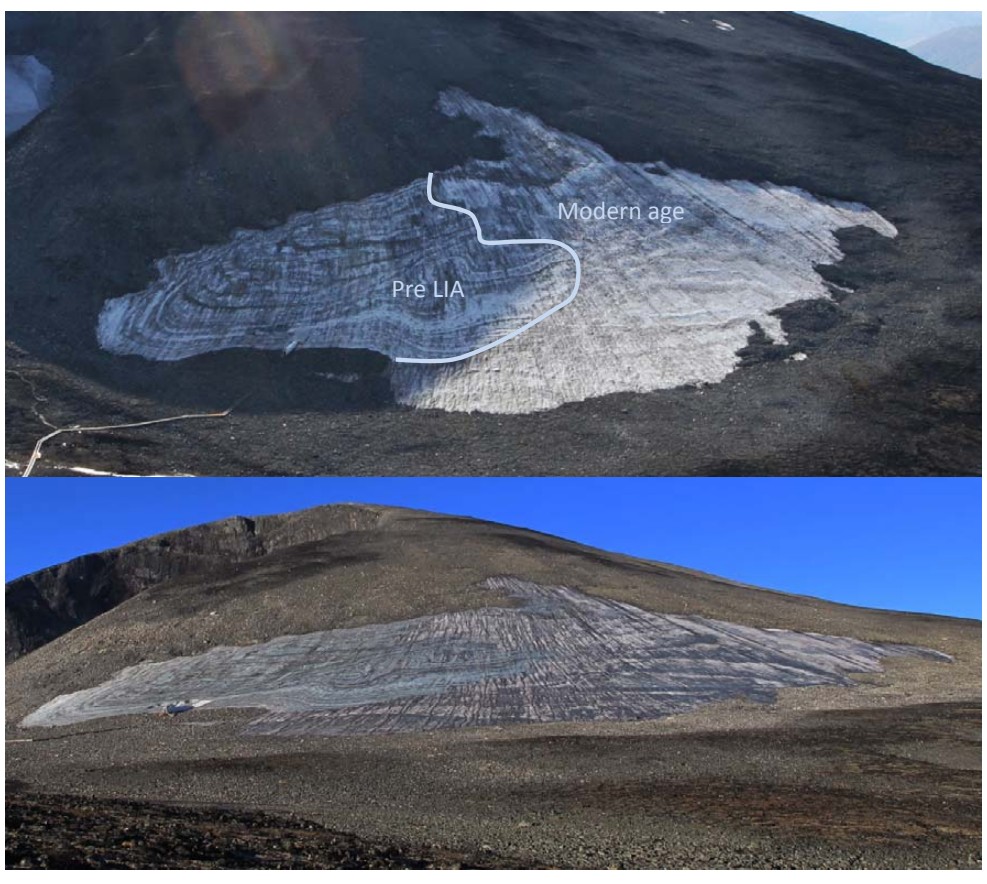

Figure 6. Photos of Juvfonne 17 September 2014 (upper) and 10 September 2014 (lower)
showing the pre 'Little Ace Age' surface exposed in central and southern parts of the ice
patch (left side). The area on Juvfonne in the north-west (right side) is interpreted to be ice of
modern age. The entrance of the ice tunnel is sitting on a small ridge that might be ice cored
(left side lower image). The collapsed 2010 tunnel is to the left of the entrance. Photo: Glacier
Archaeology Program/Oppland County Council (upper) and L. M. Andreassen (lower).



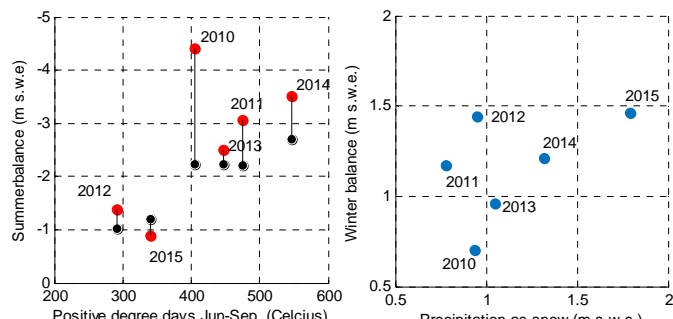

Figure 7. Summer (left) and winter (right) balance plotted against summer temperature
(positive degree days) and precipitation as snow, respectively. For the summer balance the
black markers are calculated melt using a degree-day model with typical values calibrated
from nearby glaciers (3.5 mm/°Cday for snow and 7.5 mm/°Cday for ice). Winter
precipitation is obtained from seNorge (Engeset et al., 2004).





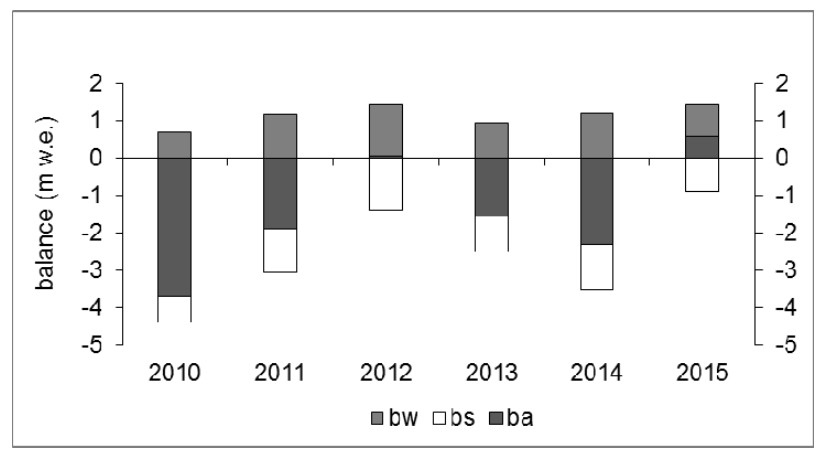

3      Figure 8. Mass balance measurements at stake J2 on Juvfonne: bw – balance winter, bs –

4      balance summer, ba – annual (net) balance. See figure 3a for position of stake.





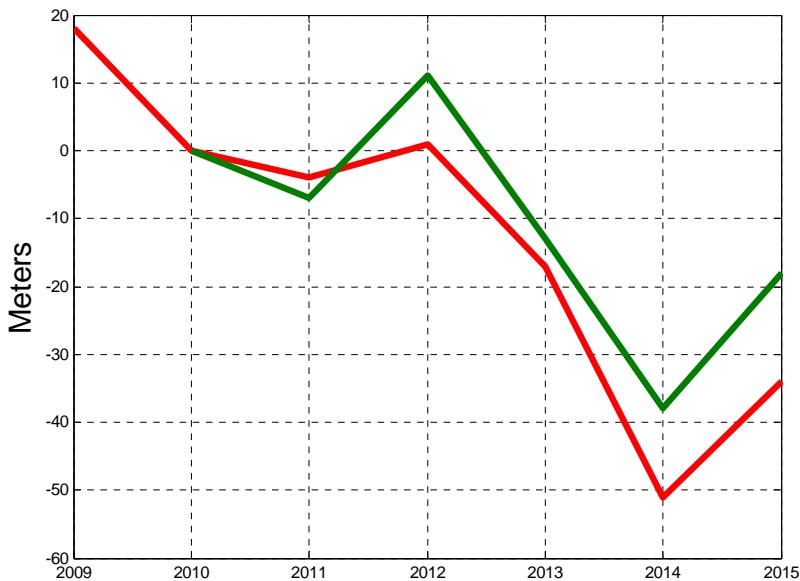

3    Figure 9. Front position of Juvfonne measured at two locations relative to the 2010-front.

4    Minima are observed in 2011 and 2014. The front retreat 2009-2014 was measured to 69 m.

5    For position of measurements, see figure 3a.  Red - JF1, Green – JF2.

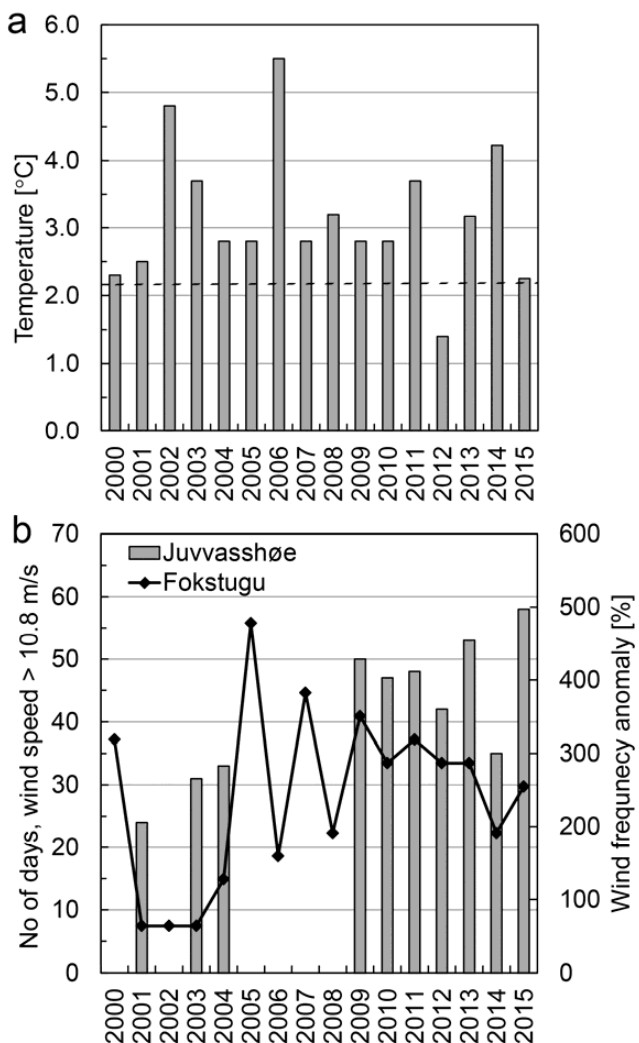

Figure 10. Meteorological data from the station at Juvvasshøe (750 m from the front of
Juvfonne) and Fokstugu 70 km NE a) Juvvasshøe June-September mean Air Temperature.
The black dotted line denotes the 1971-2000 mean, obtained from the interpolated seNorge
dataset (Engeset et al. 2004). b) Number of days for the period June-September with strong
breeze or higher (wind speed above 10.8 ms-1) at Juvvasshøe (grey bars) and at Fokstugu
(black line), the latter shown as anomaly (in %, right axes) with respect to 1971-2000 mean.





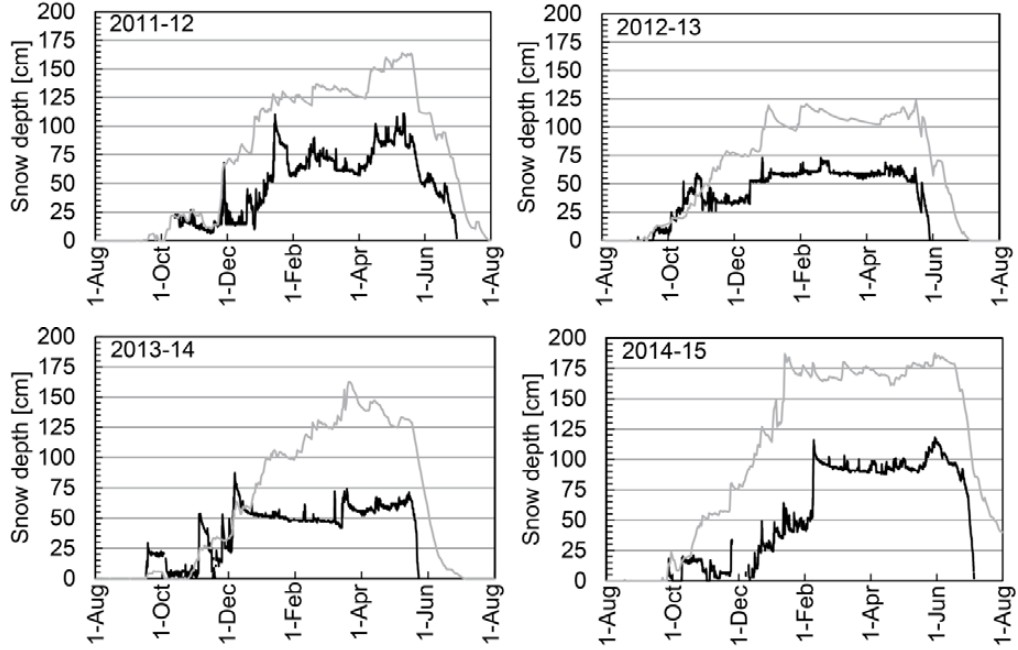

3    Figure 11. Hourly snow depth measurements (black lines) from the station 95 m from the

4    front of Juvfonne (see Figure 3a for position). Grey lines show modelled daily snow depth

5    from seNorge (Engeset et al. 2004).





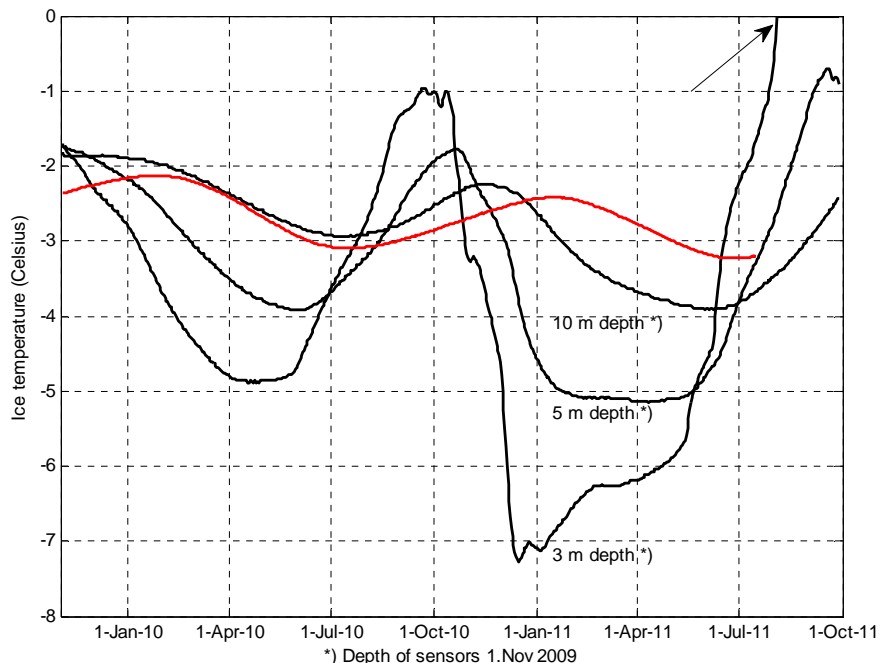

Figure 12. Temperature for November 2009-September 2011 in a 10 m deep borehole in the
Juvfonne ice patch (see Figure 3a for position). The red line is the temperature at 10 m depth
in the P31 permafrost borehole 750 m north from the ice patch (see Figure 2 for location).
Arrow points to the time when the sensor placed at 3 m depth in autumn 2009 melted out. The
entire thermistor string melted out in mid-September 2014.





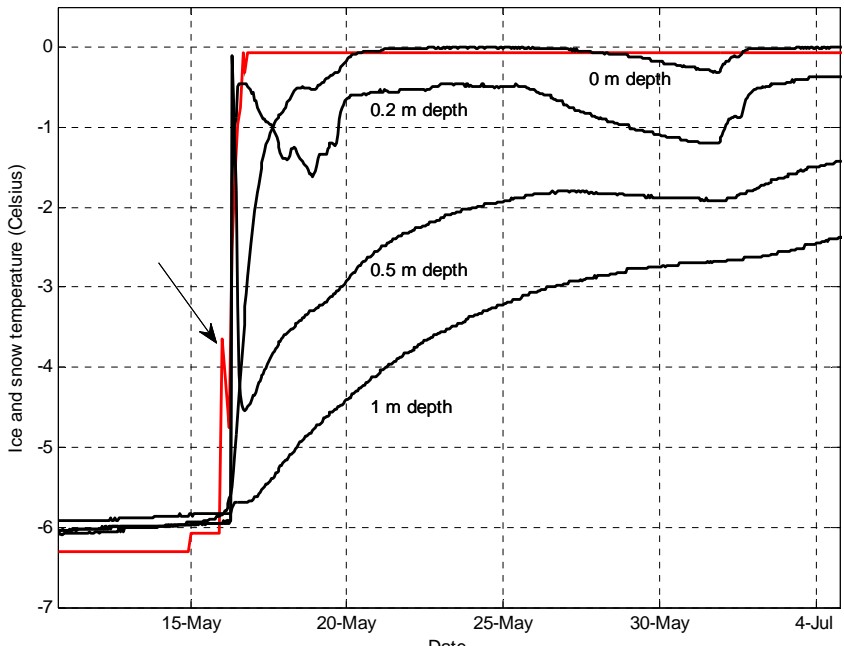

Figure 13. Plot of temperature measurements in ice and snow at the onset of thaw in May
2010 (position at the thermistor shown in figure 3a). The depth reference is the ice surface the
previous autumn. The red line is the snow temperature 0.25 m from the base of the snow
cover. The arrow point the first signal of surface meltwater refreezing close to the base of the
snow cover.





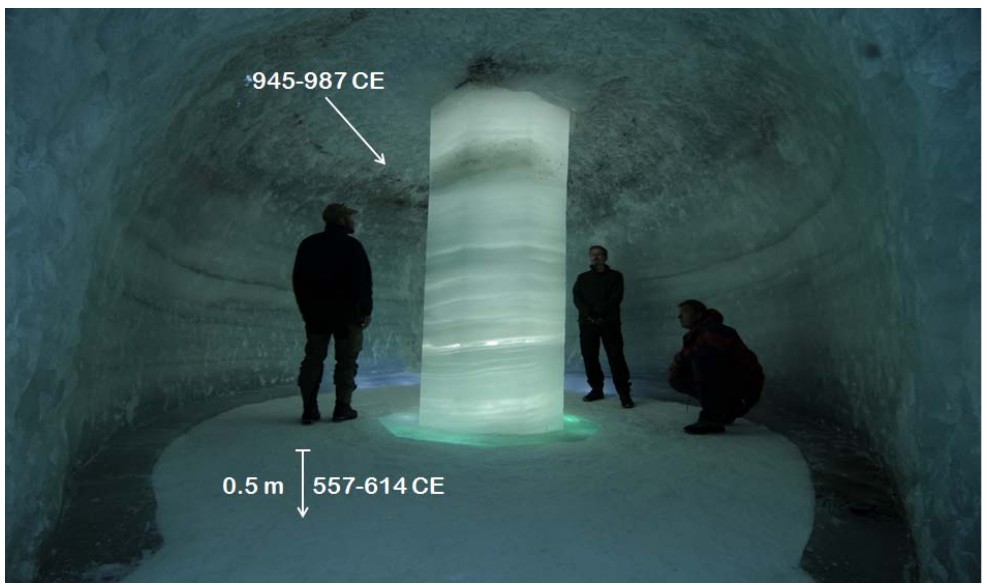

3  Figure 14. Photo from the old ice tunnel excavated in 2010 showing the layering in the ice

4  and position of two samples for radiocarbon dating. Photo: Klimapark2469 AS.

