# Peer review of "Climate change threatens archeologically significant ice"

_The Cryosphere, 2016_

## Referee Comment (RC1) · Anonymous Referee #1 · 3 Jul 2016

This paper provides an interesting analysis of the physical characteristics and recent mass balance of an ice patch in northern Norway, and provides information about a topic which has been little investigated in the past. The results are certainly interesting, but the paper is currently quite simplistic and underdeveloped compared to the rich datasets that are available for analysis. The paper basically lists the different characteristics of the ice patch, but does little to integrate them and to really explore the different processes that might be driving its temporal and spatial changes. For example, wind is stated to be an important factor in the ice patch development, but no proper analysis of the wind dataset and its connections to air temperatures and surface melt rates is

made. Similarly, no calculations are made of likely internal deformation rates for the observed ice thicknesses and surface slope. There is a considerable glaciological literature that could help with these kinds of calculations, but this is little referenced at the moment. These kinds of analyses could lift the paper from its current simplistic form to one that could really provide useful long-term insights into the factors that control ice patch growth and decline.

There is considerable duplication between the latter sections, with the Conclusions basically just providing a bulleted list of what's already been said in the Discussion and Results. The paper would also benefit from a thorough read by a native English speaker; there are currently many (generally minor) typos and language issues, some of which I detail below, but several others that I don't. Finally, several of the figures and tables could do with improvement, as detailed below.

Here are a list of comments by line number:

P2, L20: for a reader who may be unfamiliar with Otzi, please indicate where he was found

P3, L6: it would be good to add some more details about the finds at other ice patches around the world, such as the clothing associated with Otzi, spears in Yukon ice patches, etc.

P3, L13: 'differed' should be 'differentiated'

P3, L18: to help with the differentiation between glaciers and ice patches it would be useful to specify the ice thickness needed to cause ice motion (i.e., ∼40 m according to most textbooks)

P4, L6: change 'was excavated' to 'were excavated'. Also need to specify where the ice patch was that was investigated: from this para it's not even obvious that it's in Norway!

P4, L29: it would be useful to state what the ELA is on the nearby glaciers

P5, L6 (and elsewhere): there should be a space after every semi-colon. At the moment the references run into each other due to this space being missing.

P5, L19: where exactly 'in the area' were these boreholes and air temp measurements installed? I also think that you mean 'temperature sensors' rather than 'temperature measurements'

P6, L9: change 'Totally' to 'A total of'

P6, L11: please provide more information about these measurements: e.g., what was the flight altitude above the ground, what was the name of the instrument, what data was used for positioning?

P6, L18/19: some words are missing from this sentence: I think that you need to say 'were made following standard. . .'

P7, L1: please provide information on how the GNSS data was processed (e.g., using a base station, using precise point positioning?)

P7, L6: please add a label to Fig. 2 to show the location of this station

P7, L18: delete extra bracket from end of this sentence

P7, L22: it would be useful to provide some information about how the tunnels were excavated. E.g., using chainsaws? Did the excavation cause any disturbance to the surrounding ice?

P9, L5: later in the paper (P15, L8) you say that 'there are several organic/debris layers' observed within the ice tunnels. These seem to be just as likely, or perhaps more likely, to explain the layering observed in the GPR profiles.

P10, L14: this sentence makes it sound as if the ice patch almost doubled in size between 2014 and 2015 (0.101 to 0.186 km2), but based on the presence of an asterisk in Table 1 it appears that this growth was entirely due to the presence of temporary snow rather than ice. This should be made clearer in the text, and I don't believe that

it's fair to include temporary snow in the calculation of the ice patch area.

P10, L27: please state here as to what defines a 'strong breeze', and how that value was chosen

P11, L1: change 'peaks out' to 'stands out'

P11, L3-L6: there is no data presented to back up the statements in this para, so either the para should be deleted or the data should be provided.

P11, L13: I haven't heard the term abrasion used much in relation to snow events; 'wind scouring' is a more commonly used term, and would seem to be a better descriptor here.

P11, L13: change 'not take' to 'don't take'

P12, L1-4: please indicate the depth of the winter cold wave. Also please explain why the heat flow into the ice would gradually decrease during the melt season. And approximately how much superimposed ice forms each year?

P13, L1: change 'obtained results' to 'results obtained'

P13, L19: it's not clear from the text as to why 'increased accumulation towards the front of the ice patch. . . is probably a response to increased melt'. Please explain.

P13, L26-29: please provide information to back up these statements. You have the wind, temperature and ablation data, so you need to provide specific data that shows the patterns that you are arguing for.

P14, L1-3: if you make comparisons with recent major Greenland melt events you have to persuade the reader that the same conditions prevail at Juvfonne as they did in Greenland, but this isn't done at the moment.

P14, L8-9: delete 'One'. Also provide the specific date that you're referring to in this sentence (I presume that it's the storm that occurred around Feb. 5 in Fig. 11?)

P14, L10: I'm unclear as to what event you're referring to here. Please provide a specific date so that it can be connected to the patterns shown in Fig. 11

P14, L23-24: if you say that the ice patches have a similar thermal regime to nearby glaciers, then please describe what the thermal regime of the nearby glaciers actually is

P14, L29: state the ice thickness used to determine this basal shear stress

P15: L1-3: please provide reference to previously published studies that indicate the shear stress required for ice deformation to occur. There are several laboratory studies that have investigated this, so this could provide insight into the likely amount of deformation that is currently occurring, and that occurred in the past.

P15, L5: change 'theses layer' to 'these layers'

P15, L13-L16: I don't understand what the point of this para is. What are you trying to say?

P15, L21: I don't understand what 'environmental treats' are. Please define.

P16, L5: it would be good if this photo could be incorporated into this study, as it would really help to extend the timeline provided in Table 1

P17, L16: delete 'One'

Table 2: this table is poorly organized and difficult to follow, with inconsistent placing of columns between different part of the table. For example, some parts of the table have a 'Comments' column, others have a 'Dated material' column, while others have neither of these. Some sample ages are only given with 1 sigma, others are with 2 sigma. Some ages are given in relation to 1950, others are BCE. The table needs completely reworking and tidying up to make it consistent throughout.

Table 3: I don't see the value in including this table. For the (limited) information it provides it seems that it could just be incorporated into the text

Table 4: this table makes little sense by itself as from the caption it's not even possible to know what it relates to, and none of the data given in the table are really described or evaluated in the text. It should either be deleted or better described and better integrated into the manuscript.

Figure 1: this map is pretty poor quality and is missing basic information such as a scale or elevations. If you can't find better quality vector data it would be better to use something like a Landsat 8 image for the base map.

Figure 2: provide date of photo, and the direction in which the photo was taken. Also add labels to show where the P30 and P31 boreholes are located.

Figure 3: this figure needs a scale bar. Also change 'ortofoto' to 'orthophoto' in caption Figure 6/7 (and check elsewhere): use a, b, etc. to label figure parts rather than terms such as upper, lower, left and right

Figure 8: the base of the bars for 2010 and 2013 are cut off, so it's not clear what the bs values are for these years

Figures 12/13: it's very difficult to distinguish between the black lines then they cross each other. Please use a different colour (or different shade of the same colour) for each line.

Figure 14: very nice picture!

---

## Referee Comment (RC2) · Anonymous Referee #2 · 10 Jul 2016

This is a interesting research project at a very interesting site. The authors collected an impressive array of data from the perennial ice patch studied. This makes a contribution to the field as there are relatively few studies on ice patches, their development and evolution to draw information from. However, the paper lacks a central theme that ties all the data together, and more importantly, the analysis and interpretation of the data presented is rather superficial.

General comments: Overall, the paper is fairly well written but has a number of topographic and grammatical errors that, in some places, could lead to confusion. I have

identified a few of these below, but a thorough copy edit should be done. As well, the authors could have done a better job in placing their findings in the broader context. For example, a similar study from the Canadian Arctic was published a few years ago (Meulendyk, T. et al., 2012. 'Morphology and development of ice patches in Northwest Territories, Canada.' Arctic 65, 43-58). It could have been used as a comparison to delve deeper into age, development, internal structure and radar stratigraphy of the results from this study. Further, the authors collected georadar and GNSS data to image the ice thickness and bed topography, but did not do a topographic correction to the radar lines to reveal the true internal structure of the ice body. The depth of the samples for radiocarbon dating should be given and so they can be put into a proper stratigraphic context.

Specific comments: P14, L12-13. I disagree that perennial ice patches can be used as indicators of permafrost. Just like warm-based glaciers, ice patches can be at the melting point at there base with no permafrost below them.

P15, L5 change theses to these

P15, L11-12 Explain why you suggest that at other ice patches the age of the ice does not correlate to that of the organic layers.

P15, L21 change treats to threats

P16, L8-12 This paragraph is unclear. All the dating is relative as all sample could be contaminated with carbon from different times.

P16, L29 The authors refer to the ice patch not developing into a glacier with basal sliding. However, earlier they argue that it is cold based and underlain by permafrost, in which case you wouldn't expect basal sliding. See other papers on cold based glaciers. The ice temperatures and evidence of internal deformation in Figure 6 suggests that at least at some point it has been a polar style glacier (ie. cold based).

P17L17 change events to event

P17L24-29 The data presented are not detailed enough to support an assertion such as this.

P23L8-12 instead of referencing theses that are difficult to get ahold of, it would be good to reference published articles if they have come out from this work.

P25Table 2 It would be good to have the depth, or stratigraphic position, of the samples presented here to better understand the radiocarbon dates that in some cases appear to be out of order (e.g. L28&33)

P26Table 3 change intp to into

P31 Figure 4 Topographic correction should be applied to show true stratigraphic relations ships such as in Figure 5. As they are presented the unconformity in the two figures appears to be very different. As well, there seems to be a problem with the application of gain to this profile. The processing methodology is not presented in the methods section, so it is unclear what was done. However, the uniform 15 ns of muted returns above the basal reflection suggests that the gain window may have been too large or that there was some other error in the processing.

P34 Figure 7 – the winter precipitation used appears to be the modeled values estimated from the regional weather data instead of the on-site data as shown Figure 11, where the modeled data is shown to be dramatically different than the measured.

---

## Short Comment (SC1) · 18 Jul 2016

Thank you for a precise review. This feedback gives us inspiration to continue this research on ice patches in Scandinavia. The referee points to two examples that are underdeveloped compared to rich datasets available (wind and ice deformation rates). Our ambition with this paper has been to do an exploratory analysis of field data as a basis for future studies. The available wind dataset is from a meteorological station a few hundred meters from the ice patch. This dataset is unfortunately not representative for the ice patch. This is also the case for the snow accumulation measurements less

than 100 meters from the front of the ice patch. We think this illustrates challenges: spatial variability of the surface energy balance and redistribution of snow. These topics need to be addressed in detail in future research in order to do reliable calculations. We will certainly add calculations of likely ice deformation rates. Since the time perspective is thousands of years, even very small deformation rates could be significant.

---

## Author Comment (AC1) · 30 Aug 2016

**R. S. Ødegård et al.**

rune.oedegaard@ntnu.no

Comments on the general remarks from the reviewers: (Our comments are in italic)

The overall objective of this study is to do an exploratory data analysis of field data to better understand the governing processes of ice patch mass balance and Holocene development. Such an exploratory approach is normally a good research strategy when moving into new territory. The long-term objective is modelling studies to get a better quantitative understanding of the processes controlling the growth and decline of ice patches in this alpine environment. Design of models requires a basic understanding of the governing processes and how they interact. We think this study was successful to bring the state of knowledge to a level where such models can be designed. One additional dimension in this research is the cooperation with the archeologist to help them in their interpretation of finds and give some advice regarding the cultural management perspective and future development. Based on the feedback from both reviewers we have tried to clarify better the objectives (short-term, long-term) and make a better integration of the results in the conclusion. We have also made some changes in the data analysis with particular focus on the limitation of the available data (wind, mass balance) regarding quantitative calculations of turbulent fluxes, ice deformation etc. However, our intention in this study was to explore the possibilities. The quantitative modelling studies will be the next step.

New text to the introduction: "The overall objective of this study is to do an exploratory data analysis of field data to better understand the governing processes of ice patch mass balance and Holocene development. The long-term objective is to design reliable models of the growth and decline of ice patches in this alpine environment. One additional dimension in this research is the cooperation with the archaeologist to help them in their interpretation of finds and give them some advice regarding future development."

Chapter 6 – Conclusion is re-written "6. Conclusions and future perspectives"
processes that might be driving its temporal and spatial changes. For example, wind is stated to be an important factor in the ice patch development, but no proper analysis of the wind dataset and its connections to air temperatures and surface melt rates is made. Similarly, no calculations are made of likely internal deformation rates for the observed ice thicknesses and surface slope. There is a considerable glaciological literature that could help with these kinds of calculations, but this is little referenced at the moment. These kinds of analyses could lift the paper from its current simplistic form to one that could really provide useful long-term insights into the factors that control ice patch growth and decline.

Chapter 5.3 was rewritten to include calculations of deformation rates.

There is considerable duplication between the latter sections, with the Conclusions basically just providing a bulleted list of what's already been said in the Discussion and Results.

The paper would also benefit from a thorough read by a native English speaker; there are currently many (generally minor) typos and language issues, some of which I detail below, but several others that I don't. We have made some corrections in addition to those suggested by the reviewers. Otherwise we rely on the English copy-editing provided by the journal if the paper is accepted for TC.

Finally, several of the figures and tables could do with improvement, as detailed below. Here are a list of comments by line number:

P2, L20: for a reader who may be unfamiliar with Otzi, please indicate where he was Found Included in text.

P3, L6: it would be good to add some more details about the finds at other ice patches around the world, such as the clothing associated with Otzi, spears in Yukon ice patches, etc. The authors of this paper have no background in archeology. We have a short introduction with references to finds, but we don't have the background for a more detailed introduction. Based on the comments from reviewers we have added 2 references from Yukon, but it is difficult for us to make more extensive references based on the vast literature available. Added references from Yukon (Hare et al, Meulendyk et al.).

P3, L13: 'differed' should be 'differentiated' Done

P3, L18: to help with the differentiation between glaciers and ice patches it would be useful to specify the ice thickness needed to cause ice motion (i.e., _40 m according to most textbooks) Chapter 5.3 is rewritten including calculations of ice deformation.

P4, L6: change 'was excavated' to 'were excavated'. Also need to specify where the ice patch was that was investigated: from this para it's not even obvious that it's in Norway! Done

P4, L29: it would be useful to state what the ELA is on the nearby glaciers ELA added. "The ELA increases with distance from coast from 1780 m a.s.l. at Storbreen to 2150 m a.s.l. at Gråsubreen (Kjøllmoen et al., 2011)"

P5, L6 (and elsewhere): there should be a space after every semi-colon. At the moment the references run into each other due to this space being missing. An update of the output style fixed the problem.

P5, L19: where exactly 'in the area' were these boreholes and air temp measurements installed? I also think that you mean 'temperature sensors' rather than 'temperature measurements' New text: "In 2008 an altitudinal transect of boreholes and adjacent air temperature sensors were installed at three sites ranging from shallow seasonal frost to permafrost"

P6, L9: change 'Totally' to 'A total of' Done

P6, L11: please provide more information about these measurements: e.g., what was the flight altitude above the ground, what was the name of the instrument, what data was used for positioning? New text: "The ice patch and surrounding terrain was scanned with an air-borne laser (Leica ALS70) on 17 September 2011. The company COWI AS, on assignment from Norwegian Water Resources and Energy Directorate, carried out the laser scanning and the processing of the data. The flight altitude was 10100-11800 feet (3078-3597 m a.s.l). The area was scanned with 5 points m-2. Quality controls and accuracy assessments revealed an accuracy better than 0.1 m in surface elevation. Aerial photos were taken on the same day. These data were used to produce a high quality DTM and orthophotos of the ice patch surface and surrounding terrain. The DTM was resampled to a resolution of 1 m."

P6, L18/19: some words are missing from this sentence: I think that you need to say 'were made following standard' Done

P7, L1: please provide information on how the GNSS data was processed (e.g., using a base station, using precise point positioning?) Text added: The extent of the Juvfonne ice patch has been surveyed by foot with GNSS with a Topcon receiver mounted on a back pack and one reference receiver mounted in a fixed base point (Fig 3a, Table 1). The GNSS data was processed with Topcon software TTOOLS version 8. '

P7, L6: please add a label to Fig. 2 to show the location of this station Location added on figure.

P7, L18: delete extra bracket from end of this sentence The Norwegian Mapping Authority Done

P7, L22: it would be useful to provide some information about how the tunnels were excavated. E.g., using chainsaws? Did the excavation cause any disturbance to the surrounding ice? New text: "The tunnels were excavated with specially designed ice axes causing minimal disturbance to the surrounding ice. The tunnels gave an excellent opportunity to verify the radar data and to collect organic material and ice for radiocarbon dating"

P9, L5: later in the paper (P15, L8) you say that 'there are several organic/debris layers'

observed within the ice tunnels. These seem to be just as likely, or perhaps more likely, to explain the layering observed in the GPR profiles. From observations in the tunnels the organic layers are discontinuous. New text: "The bed reflection was clearly seen in the radar plots (see example in Fig. 4). In addition the ice layering was detected on most of the plots, probably due to density differences in the ice layers (air bubbles) (Hamran et al., 2009) or organic layers."

P10, L14: this sentence makes it sound as if the ice patch almost doubled in size between 2014 and 2015 (0.101 to 0.186 km2), but based on the presence of an asterisk in Table 1 it appears that this growth was entirely due to the presence of temporary snow rather than ice. This should be made clearer in the text, and I don't believe that it's fair to include temporary snow in the calculation of the ice patch area. Added text: "Furthermore, observations in field show that the ice is very thin along the margins. In 2015, seasonal snow covered the entire margin, and the measured area of 0.186 km2 is thus only to be considered a maximum extent, not the actual ice patch area. "

P10, L27: please state here as to what defines a 'strong breeze', and how that value was chosen The definition was written in the Figure caption for Figure 10b (P37, L7) and follow the international classification given by World Meteorological Organization and is now also included in the text (see below). The available wind dataset is from Juvvasshøe, located 750 meters from the ice patch, and from Fokstugu, 70 km NE of Juvasshøe. The wind speed at Juvvasshøe and Fokstugu is unfortunately not representative for the ice patch. Experience gained through field work at Juvfonne suggests that the wind speed is only 10 to 50% compared to Juvvasshøe, especially during prevailing westerly winds. Thus strong breeze observed at the two meteorological stations was used as a lower limit to get sufficient high wind speeds for effective turbulent fluxes at Juvfonne.

The text was changed to: "Due to the sheltered setting of Juvfonne compared to the meteorological stations, strong breeze (wind speed above 10.8 ms-1) was used as a lower limit to get sufficient high wind speeds for effective and enhanced turbulent fluxes at Juvfonne. In general there is a high frequency (35-58 days per season) of strong breeze during the period 2009-2015 (Fig. 10b)." According to this our text at P7, L6-7 was also changed: "It is the highest official meteorological station in Norway and is freely exposed and representative for this study, except for wind speed."

P11, L1: change 'peaks out' to 'stands out' Done

P11, L3-L6: there is no data presented to back up the statements in this para, so either the para should be deleted or the data should be provided.

Snow accumulation and erosion are among the most discussed processes in context with local wind speed variations in mountainous areas (see e.g. Liston and Sturm 1988; Lehning et al. 2007; Dadic et al. 2009). Data is now provided with a new figure included (Figure 11).

The text was changed to: "For snow accumulation or abrasion on ice patches wind speed and wind direction is crucial (Lehning et al. 2008; Dadic et al. 2010). There are great variations from year to year in respect to frequency of strong gale and wind direction. During the two stormiest winters 2011-12 and 2013-14, the frequency of strong gale was 15.7 % and 17.3 %, respectively (Figure 11)."

Lehning M, Löwe H, Ryser M, Raderschall N. Inhomogeneous precipitation distribution and snow transport in steep terrain. Water Resour Res 44(7), 10.1029/2007WR006545. Dadic R, Mott R, Lehning M, Burlando P. Wind influenceon-snow depth distribution and accumulation over glaciers. J Geophys Res 115 (F01012), 10.1029/2009JF001261.

New figure text: Figure11. Relative frequency (as % of all hourly observations) of strong gale or more ($\geq$ 20.8 ms-1) at Juvvasshøe during winter (Oct-Apr) 2009-2015 for the wind sectors SE to NW. The values inserted show the total frequency of strong gale or more.

P11, L13: I haven't heard the term abrasion used much in relation to snow events; 'wind scouring' is a more commonly used term, and would seem to be a better descriptor here. Done

P11, L13: change 'not take' to 'don't take' Done

P12, L1-4: please indicate the depth of the winter cold wave. Also please explain why the heat flow into the ice would gradually decrease during the melt season. And approximately how much superimposed ice forms each year? Winter cold wave is a confusing expression here since there is cold ice below the level of meltwater percolation. Paragraph has been rewritten: "There is cold ice below the level of meltwater percolation, which means that the heat flow into the ice is gradually decreasing during the melt season. Because of this heat flow superimposed ice forms at the level of impermeable ice, generally less than 0.1 m."

P13, L1: change 'obtained results' to 'results obtained' Done

P13, L19: it's not clear from the text as to why 'increased accumulation towards the front of the ice patch probably a response to increased melt'. Please explain.

Added at the end of the sentence: "which will increase the snow accumulation at the leeward side of prevailing westerly winds".

P13, L26-29: please provide information to back up these statements. You have the wind, temperature and ablation data, so you need to provide specific data that shows the patterns that you are arguing for. We have only one ablation stake that survived the measurement period. For the asymmetric melting we have to rely on field observations reporting extreme melt in early-mid August 2010 and pictures. The table below shows the warmest 10-day periods each year. 8-18 August was the warmest in 2010 with average wind speed 3.4 m/s, humidity 79.5% and wind direction from SW. The wind speeds are not representative for Juvfonne, but SW is an exposed wind direction for Juvfonne. Table below show median values of wind speed, air temperature, relative humidity and wind direction of the warmest 10-day period during Jun-Jul-Aug each

year. 8-18 August 2010 is a period with high wind speeds, high humidity and most important median wind from SE. Wind speed [ms-1] Temperature [°C] Humidity [%] Wind direction [°] Ending date for 10-day period 2009 2.3 11.1 59.5 192.0 2009-07-04 2010 3.4 7.8 79.5 139.0 2010-08-18 2011 2.6 8.7 81.5 183.0 2011-08-04 2012 2.5 6.5 77.0 155.0 2012-08-20 2013 2.5 9.4 65.5 256.0 2013-07-29 2014 2.7 11.0 67.5 182.5 2014-07-28 2015 7.3 7.8 53.0 162.0 2015-08-23

Added text: "Extreme melt was reported in early-mid August. The warmest 10-day period in 2010 was 8-18 August. Average wind speed was 3.4 m/s from SE (humidity 79.5%)."

P14, L1-3: if you make comparisons with recent major Greenland melt events you have to persuade the reader that the same conditions prevail at Juvfonne as they did in Greenland, but this isn't done at the moment. The comparisons with Greenland were meant to highlight situations that lead to a significant increase in nonradiative energy fluxes and the importance of exposure to wind. A similar exceptional melt event caused by a warm, very humid storm system in the Central Cascade Mountains of Oregon was reported by Marks et al. 1998. They showed that the snow melt were enhanced by strong wind, high air temperature and high humidity. At higher unsheltered sites 60-90% of the energy for snowmelt came from sensible and latent heat exchanges, while it was only about 35% at more sheltered sites (Marks et al. 1998).

The text was changed to: "Exceptionally large melt episodes have been reported from the Central Cascade Mountains of Oregon where snow melt were enhanced by strong wind, high air temperature and high humidity (Marks et al. 1998). At higher unsheltered sites 60-90% of the energy for snowmelt came from sensible and latent heat exchanges, while it was only about 35% at more sheltered sites (Marks et al. 1998). Recently similar extreme melt events have been reported from the southern and western part of Greenland ice sheet in July 2012, where nonradiative energy fluxes (sensible, latent, rain, and subsurface collectively) dominated the ablation area surface energy budget during multiday episodes (Fausto et al., 2016)."

Added reference: Marks D, Kimball J,Tingey D, Link T. The sensitivity of snowmelt processes to climate conditions and forest cover during rain-on-snow: a case study of the 1996 Pacific Northwest flood. Hydrol Process 1998; 12: 1569–1587.

P14, L8-9: delete 'One'. Also provide the specific date that you're referring to in this sentence (I presume that it's the storm that occurred around Feb. 5 in Fig. 11?) Changed to: "Single storm events with westerly winds could account for almost 50% of the winter accumulation in less than 24 hours, like the storm February 7-8 in 2015 (Figure 11, 2014-15)."

P14, L10: I'm unclear as to what event you're referring to here. Please provide a specific date so that it can be connected to the patterns shown in Fig. 11 Changed to: "Spring snow accumulation with insignificant wind drift could also influence mass balance, like the period from early April to mid May 2012 where more than 40 cm of snow accumulated (Figure 11, 2011-2012)."

P14, L23-24: if you say that the ice patches have a similar thermal regime to nearby glaciers, then please describe what the thermal regime of the nearby glaciers actually is New text: The temperature measurements at Juvfonne show that there is sufficient melt water to bring the permeable snowpack to an isothermal condition within a few weeks in early summer (Fig. 13). Below the seasonal snowpack, the ice remains cold during the summer with temperatures on the range -2 - -4°C at 5-10 m depth (Fig. 13). In Norway most glaciers are considered to be temperate, although measurements are available for only a few glaciers (Andreassen and Winsvold, 2012). Recent observations from nearby glaciers in Jotunheimen, reveal that at the lower parts of Storbreen the winter cold wave is removed during summer, but remained at Hellstugubreen and Gråsubreen (Sørdal, 2013;Tachon, 2015). The temperature measured close to the equilibrium line at Hellstugubreen (-1°C) and Gråsubreen (-2°C) were warmer than the temperature measured at similar depths at Juvfonne (-3°C).

P14, L29: state the ice thickness used to determine this basal shear stress Chapter

5.3 is rewritten including ice thickness.

P15: L1-3: please provide reference to previously published studies that indicate the shear stress required for ice deformation to occur. There are several laboratory studies that have investigated this, so this could provide insight into the likely amount of deformation that is currently occurring, and that occurred in the past. Chapter 5.3 is rewritten including references.

P15, L5: change 'theses layer' to 'these layers' Done

P15, L13-L16: I don't understand what the point of this para is. What are you trying to say? Deleted.

P15, L21: I don't understand what 'environmental treats' are. Please define. Spelling error corrected

P16, L5: it would be good if this photo could be incorporated into this study, as it would really help to extend the timeline provided in Table 1 New figure 17 with old photo. Figure text: Figure 17. Picture taken from Vesljuvbrea towards north-northwest showing Juvfonne from around 1900. The surface slope of Juvfonne is estimated to approximately 15°. Height and length estimate from map based on position in the picture. The upper and northern part of Juvfonne is not seen on the picture.

P17, L16: delete 'One' Done

Table 2: this table is poorly organized and difficult to follow, with inconsistent placing of columns between different part of the table. For example, some parts of the table have a 'Comments' column, others have a 'Dated material' column, while others have neither of these. Some sample ages are only given with 1 sigma, others are with 2 sigma. Some ages are given in relation to 1950, others are BCE. The table needs completely reworking and tidying up to make it consistent throughout. New table 2 is totally reorganized. All dates from Juvfonne changed to BP in the manuscript.

Table 3: I don't see the value in including this table. For the (limited) information it provides it seems that it could just be incorporated into the text Deleted.

Table 4: this table makes little sense by itself as from the caption it's not even possible to know what it relates to, and none of the data given in the table are really described or evaluated in the text. It should either be deleted or better described and better integrated into the manuscript. Deleted.

Figure 1: this map is pretty poor quality and is missing basic information such as a scale or elevations. If you can't find better quality vector data it would be better to use something like a Landsat 8 image for the base map. New figure 2 with a simple map. We have plenty of available vector data, but decided to keep it simple.

Figure 2: provide date of photo, and the direction in which the photo was taken. Also add labels to show where the P30 and P31 boreholes are located. Date of photo not available (month and year inserted). The rest is corrected.

Figure 3: this figure needs a scale bar. Also change 'ortofoto' to 'orthophoto' in caption With the UTM references in meters a scale bar is not included.

Figure 6/7 (and check elsewhere): use a, b, etc. to label figure parts rather than terms such as upper, lower, left and right Done Figure 8: the base of the bars for 2010 and 2013 are cut off, so it's not clear what the bs values are for these years OK in Word-version. Problem in PDF-version

Figures 12/13: it's very difficult to distinguish between the black lines then they cross each other. Please use a different colour (or different shade of the same colour) for each line. New figures with different colors.

Figure 14: very nice picture OK – text on photo changed to BP
lected an impressive array of data from the perennial ice patch studied. This makes a contribution to the field as there are relatively few studies on ice patches, their development and evolution to draw information from. However, the paper lacks a central theme that ties all the data together, and more importantly, the analysis and interpretation of the data presented is rather superficial. General comments: Overall, the paper is fairly well written but has a number of topographic and grammatical errors that, in some places, could lead to confusion. I have identified a few of these below, but a thorough copy edit should be done. As well, the authors could have done a better job in placing their findings in the broader context. For example, a similar study from the Canadian Arctic was published a few years ago (Meulendyk, T. et al., 2012. 'Morphology and development of ice patches in Northwest Territories, Canada.' Arctic 65, 43-58). Reference included. The authors of this paper have no background in archeology. We have a short introduction with references to finds, but we don't have the background for a more detailed discussion of finds. It could have been used as a comparison to delve deeper into age, development, internal structure and radar stratigraphy of the results from this study. Further, the authors collected georadar and GNSS data to image the ice thickness and bed topography, but did not do a topographic correction to the radar lines to reveal the true internal structure of the ice body.

New figure figure 7 with topographic corrections..

The depth of the samples for radiocarbon dating should be given and so they can be put into a proper stratigraphic context.

New figure 16: age/vertical distance to bed.

Specific comments: P14, L12-13. I disagree that perennial ice patches can be used as indicators of permafrost. Just like warm-based glaciers, ice patches can be at the melting point at there base with no permafrost below them.

Very interesting comment – no changes made to the text but we gladly included parts of the discussion below– 2 references added , Imhof 1996 and Kneisel 1998. Mountain permafrost researchers have used perennial snow patches as an indicator of permafrost. Some authors (Imhof 1996) consider perennial snow patches as permafrost by definition with a statement: "The only exception are perennial snow patches, which - by definition - cover permafrost and which are easily detectable by aerial photographs: below snow patches, the ground surface temperature cannot rise above zero degrees during the whole season." Other authors like Kneisel, 1998 use statements like "perennial snow patches as indicator of mountain permafrost". To our knowledge these types of statements have not caused any big controversy.

There is no doubt that temperate ice can survive for some years, maybe decades in a perennial snow/ice patches during an initial fast build up. However, ice patches are by definition areas with close to zero mass balance. Snow could accumulate fast and reduce heat loss to the atmosphere during most of the winter. The critical phase occurs in late autumn/early winter when cold weather occurs before the first snowfall. In summer/summers with negative balance, ice is often exposed and there is a cooling of surface ice. This is similar to the situation close to ELA of glaciers. This cooling occurs when the ice patch is at its minimum. Depending on the melt the following years, there is plenty of time (years or decades) for the cold wave to penetrate and eventually reach the base. Unlike glaciers this ice is not likely to melt because there is no movement and close to zero mass balance. When the ice is cold and stagnant, there is no way to bring it back to temperate ice. The possibility of melt at the base is another aspect that needs to be considered for an ice patch with no permafrost beneath. If the ice at the base is at the pressure melting point heat flow from below will cause basal melting. Even the geothermal heat flow in Southern Norway (50-60 mW/m2) will cause a melting of 5-6 m/years*1000. Additional heat sources like ground water are likely. With no permafrost the old ice at the base will not survive. Even 100 years with no permafrost could cause significant basal melt. The oldest ice samples at Juvfonne are within 0.5 meters of the base.

P15, L5 change theses to these Done

P15, L11-12 Explain why you suggest that at other ice patches the age of the ice does not correlate to that of the organic layers.

Se chapter 5.3 (re-written) New text: "This is necessarily not the case at other ice patches, where organic material exposed at the surface could be contaminated by surface processes or microbial activity."

P15, L21 change treats to threats Done

P16, L8-12 This paragraph is unclear. All the dating is relative as all sample could be contaminated with carbon from different times. Text added: "Contamination is not likely in the clear ice samples, which gives confidence in the dating of the ice stratigraphy."

P16, L29 The authors refer to the ice patch not developing into a glacier with basal sliding. However, earlier they argue that it is cold based and underlain by permafrost, in which case you wouldn't expect basal sliding. See other papers on cold based glaciers. The ice temperatures and evidence of internal deformation in Figure 6 suggests that at least at some point it has been a polar style glacier (ie. cold based). Chapter 5.3 rewritten in an attempt to clarify. We definitely agree that at some point this was a cold based glacier.

P17L17 change events to event Done

P17L24-29 The data presented are not detailed enough to support an assertion such as this.

Chapter 5.3 rewritten and conclusion modified. "Even a thin ice patch like Juvfonne (<20 m thick) ice deformation on Holocene time scale could be a critical factor in the interpretation of the ice layering and makes it difficult to relate the present thickness and slope of theses layer to previous thickness of the ice patch."

P23L8-12 instead of referencing theses that are difficult to get ahold of, it would be There are no papers from these theses. See also our response to P14, L23-24.
P25Table 2 It would be good to have the depth, or stratigraphic position, of the samples presented here to better understand the radiocarbon dates that in some cases appear to be out of order (e.g. L28&33)

New figure 16: age/vertical distance to bed.

P26Table 3 change intp to into Done.

P31 Figure 4 Topographic correction should be applied to show true stratigraphic relations ships such as in Figure 5. As they are presented the unconformity in the two figures appears to be very different. As well, there seems to be a problem with the application of gain to this profile. The processing methodology is not presented in the methods section, so it is unclear what was done. However, the uniform 15 ns of muted returns above the basal reflection suggests that the gain window may have been too large or that there was some other error in the processing. New figure 4 with topographic correction. Gain has been changed.

P34 Figure 7 – the winter precipitation used appears to be the modeled values estimated from the regional weather data instead of the on-site data as shown Figure 11, where the modeled data is shown to be dramatically different than the measured. Data from SeNorge are the best data for precipitation in Norway (they are modelled but based on observations).

Please also note the supplement to this comment:
http://www.the-cryosphere-discuss.net/tc-2016-94/tc-2016-94-AC1-supplement.pdf

**Supplement:**

[revised manuscript text omitted]
. Ages obtained by radiocarbon dating of clear ice and organic remains collected in the ice tunnels and from the ice patch surface. Ice samples were collected as blocks and subdivided in several sub-samples. Therefore an average value is shown for every block (JUV1, JUV2 and JUV3) except for JUV0, because JUV0_1 and JUV0_2 were taken adjacent to the plant fragment layer, dated 6600 cal BP (Poz-56955), while samples from JUV0_3 to

JUV0_8 were collected at the bottom of the wall, a few cm below the plant fragment layer.

Thus JUV0_A is the average of JUV0_1 and JUV0_2, while the other six samples were averaged as JUV0_B. Individual calibrated ages for ice sub-samples are not shown because derived ages were combined using the function in OxCal v4.2.4. ([14]C date combination).

Calibrated ages are given in years before present (cal BP, with BP = 1950) as median probability and 1 σ uncertainty range.

[revised manuscript text omitted]

balance summer, ba – annual (net) balance (. See figure Fig. 3a for position of stake).

[Figure]

Figure 9. Front position of Juvfonne measured at two locations relative to the 2010-front.

Minima are observed in 2011 and 2014. The front retreat 2009-2014 was measured to 69 m.

For position of measurements, see figure Fig. 3a. Red - JF1, Green – JF2.

[Figure]

Figure 10. Meteorological data from the station at Juvvasshøe (750 m from the front of Juvfonne) and Fokstugu 70 km NE a) Juvvasshøe June-September mean Air Temperature. The black dotted line denotes the 1971-2000 mean, obtained from the interpolated seNorge dataset (Engeset et al. 2004). b) Number of days for the period June-September with strong breeze or higher (wind speed above 10.8 ms-1) at Juvvasshøe (grey bars) and at Fokstugu (black line), the latter shown as anomaly (in %, right axes) with respect to 1971-2000 mean.

[Figure]

Figure 11. Relative frequency (as percentage of all hourly observations) of strong gale or more (≥ 20.8 ms-1) at Juvvasshøe during winter (Oct-Apr) 2009-2015 for the wind sectors SE

to NW. The values inserted show the total frequency of strong gale or more.

[Figure]

Figure 12. Hourly snow depth measurements (black lines) from the station 95 m from the front of Juvfonne (see Figure 3a for position). Grey lines show modelled daily snow depth from seNorge (Engeset et al. 2004).

[Figure]

[Figure]

Figure 13. Temperature for November 2009-September 2011 in a 10 m deep borehole in the Juvfonne ice patch (see Fig. 3a for position). The red line is the temperature at 10 m depth in the P31 permafrost borehole 750 m north from the ice patch (see Fig. 2 for location). Arrow points to the time when the sensor placed at 3 m depth in autumn 2009 melted out. The entire thermistor string melted out in mid-September 2014.

[Figure]

[Figure]

Figure 14. Plot of temperature measurements in ice and snow at the onset of thaw in May 2010 (position at the thermistor shown in Fig. 3a). The depth reference is the ice surface the previous autumn. The red line is the snow temperature 0.25 m from the base of the snow cover. The arrow point the first signal of surface meltwater refreezing close to the base of the snow cover.

[Figure]

Figure 154. Photo from the old ice tunnel excavated in 2010 showing the layering in the ice and position of two samples for radiocarbon dating. Photo: Klimapark2469 AS.

[Figure]

Figure 16. Plot of the samples in Table 2 except samples with modern age. In the inner parts of the 2012 tunnel the bed is partly exposed, which gives good distance to bed estimates. In the 2010 tunnel, the distance estimates depend on the radar data (the old tunnel partly melted out). The horizontal distance between the samples are up to 50 m.

[Figure]

Figure 17. Picture taken from Vesljuvbrea towards NNW, showing Juvfonne from around 1900. The surface slope of Juvfonne is approximately 15°. Height and length estimate from map based on position in the picture. The upper and northern part of Juvfonne cannot be seen on the picture.

---

## Referee Report (RR1)

P2, L5-13: The abstract is better than before, but I find it overly long and a bit lacking in details towards the end. I would therefore suggest deleting the last few sentences, which describe little in terms of concrete results:

"*The cumulative deformation of ice over millennia could explain the observed curved layering in the basal parts of the ice patch, which makes it difficult to relate the present thickness to previous thickness of the ice patch. Ice deformation and surface processes (i.e. wind and melt water) may have caused significant displacement of artefacts from their original position. Thus, the dating and position of artefacts cannot be used directly to reconstruct previous ice patch extent. In the perspective of surface energy and mass balance, ice patches are in the transition zone between permafrost terrain and glaciers. Future research will need to carefully address this interaction to build reliable models.*"

P3, L21: change 'is fast' to 'occurs rapidly'

P3, L22: I would put a paragraph break before 'Ice patches'

P3, L23: change ' the thermal regime' to 'their thermal regime'

P4, L9-11: The meaning of the wording 'cooperation with the archaeologist…' isn't very clear, so I would recommend rewriting or deleting this sentence

P6, L16: you use the term 'artefacts' to refer to archaeological artefacts earlier in the paper, so to avoid any ambiguity it would be better to use a different word here (e.g., 'errors'? 'problems'?)

P6, L18: to make it clear what you're referring to here I would add the word 'positional' – e.g., 'estimated positional standard deviation…'

P6, L25: should be '5 points m$^{-2}$'

P7, L12/13: provide proper references (e.g., map sheet number, publisher) for the 'topographical maps from the Norwegian mapping authorities'

P7, L15: please define what 'N50' is referring to here

P7, L17: change 'on the range 10-20 cm giving…' to 'in the range 0.1-0.2 m, giving…'

P7, L21: please explain why it is not representative for wind speed

P10, L12: wording is currently a bit unclear, so I would recommend changing it to: 'The total measured mass loss was >10 m of ice at the site…'

P10, L30: change to 'observations in the field'

P11, L12: change '10.8 ms-1' to '10.8 m s$^{-1}$'

P11, L21: please define the wind speed that 'strong gale' refers to

P12, L17: change 'the ice gradually decreasing…' to 'the ice which gradually deceases…'

P12, L19: it is ambiguous as to whether the 0.1m/year (which should be written '0.1 m yr$^{-1}$') refers to the thickness of superimposed ice or the level of impermeable ice. Please clarify.

P16, L2: change 'surface' to 'surfaces'

P16, 12-20: it's very useful to see these calculations of suggested ice deformation rates!

P16, L21: change 'calculations are uncertain' to 'calculations have high uncertainty'

P18, L5: change 'to level' to 'to a level'

P18, L29-30: this sentence is unclear, so please reword: "Since the surface ice shows modern age artefacts melted out in front of Juvfonne since 2009 have been sub-aerially exposed after the LIA but prior to 2009."

P19, L6-7: the 'well-known mass balance feedback mechanisms' may not be well known for non-experts, so please describe them here

---

## Author Response (AR2)

Final comments from the authors:

Thank you for precise feedback on the last version of the manuscript. With a few exceptions (commented below), we have changed the manuscript according the suggestions from the editor/reviewers. Enclosed a marked document with all the changes marked in red.

Comments to the changes (minor corrections are not commented):

Page 2, line 5. Deleting the last part makes the abstract more focused.

Page 3, line 31. Deleted the addition reference to Morris, 1989. Not important in this context.

Page 4, paragraph starting with line 6.

Your suggestion was "explain or delete". We decided to delete to clarify the objective.

Page 6, line 18.

Your suggestion "position standard deviaton" changed to "standard deviation in bed elevation"

Page 6, line 25.

Original text: "The flight altitude was 10100-11800 feet (3078-3597 m a.s.l.)." changed to "The flight altitude was 3080-3600 m a.s.l.."  The altitude is above sea level.

Page 7, line 13.

Original text: "Areal extent was also determined by digitising outlines from orthophotos from 2011 and from topographical maps from the Norwegian mapping authorities in 1981 and 2004." changed to "Areal extent was also determined by digitising outlines from orthophotos from 2011 and from topographical maps in 1981 and 2004 (map product coded N50 from the Norwegian mapping authorities)."

Page 7, line 22.

Original text:" It is the highest official meteorological station in Norway, and is freely exposed and representative for this study, except for wind speed." Changed to:

"Juvvasshøe is the highest official meteorological station in Norway, and is freely exposed and considered representative for this study. Due to the sheltered setting of Juvfonne compared to Juvvasshøe the wind speed at Juvvasshøe is generally higher. However, wind data from Juvvasshøe gives important information on wind direction and wind speed relevant for snow accumulation and ablation processes at Juvfonne."

Page 11, paragraph starting at line 11.

Breeze or gale as relevant cutoff?  We think we have made it clear that the cutoff for strong breeze is used related to turbulent fluxes and cutoff for strong gale is used for snow accumulation and abrasion. No changes made to the manuscript.

Page 13, line 8,9 and 14.

Minor changes to the calibration of the radiocarbon dates. The reason for these changes is that the Villigen-group has been running a parallel review in TC about the methodology used in the calibration (number of digits used in the Oxcal calculation). This manuscript was accepted last week. To ensure consistency between the publications we decided to make these minor changes. Table 2 is also changed.

Page 15, line 10.  "subsurface collectively" is deleted. Not important in this context. The term is used in the referenced paper about subsurface fluxes.

Page 16, line 10.

We think we have explained this since contamination is not likely in the clear ice samples. No changes made to the manuscript.

Page 18, line 13.

[revised manuscript text omitted]

---

## Author Response (AR3)

Authors' response:

No changes to the text.

Graphical editing of the figures:

Figure 4:  More visible arrow

Figure 7:  Text of axes - brackets changed from () to []

Figure 9: Text of x-axis changes from 'Meters' to 'Front position change [m]'

      Color of line changed from green-red to blue-red

Figure 13: Text of axes - brackets changed from () to []

Figure 14: Text of axes - brackets changed from () to []

Figure 16: Text of axes - brackets changed from () to []

      Added grid.

Figure 17: Increased font size and line width.